# Laplacian Eigenspaces, Horocycles and Neuron Models on Hyperbolic Spaces

## Abstract

We use hyperbolic Poisson kernel to construct the horocycle neuron model on hyperbolic spaces, which is a spectral generalization of the classical neuron model. We prove a universal approximation theorem for horocycle neurons. As a corollary, we obtain a state-of-the-art result on the expressivity of $f_{a,p}^1$, which is used in the hyperbolic multiple linear regression. Our experiments get state-of-the-art results on the Poincare-embedding subtree classification task and the classification accuracy of the two-dimensional visualization of images.

## 1 Introduction

Conventional deep network techniques attempt to use architecture based on compositions of simple functions to learn representations of Euclidean data (LeCun et al., 2015). They have achieved remarkable successes in a wide range of applications (Hinton et al., 2012; He et al., 2016). Geometric deep learning, a niche field that has caught the attention of many authors, attempts to generalize conventional learning techniques to non-Euclidean spaces (Bronstein et al., 2017; Monti et al., 2017).

There has been growing interest in using hyperbolic spaces in machine learning tasks because they are well-suited for tree-like data representation (Ontrup & Ritter, 2005; Alanis-Lobato et al., 2016; Nickel & Kiela, 2017; Chamberlain et al., 2018; Nickel & Kiela, 2018; Sala et al., 2018; Ganea et al., 2018b; Tifrea et al., 2019; Chami et al., 2019; Liu et al., 2019; Balazevic et al., 2019; Yu & Sa, 2019; Gulcehre et al., 2019; Law et al., 2019). Many authors have introduced hyperbolic analogs of classical learning tools (Ganea et al., 2018a; Cho et al., 2019; Nagano et al., 2019; Grattarola et al., 2019; Mathieu et al., 2019; Ovinnikov, 2020; Khrulkov et al., 2020; Shimizu et al., 2020).

Spectral methods are successful in machine learning, from nonlinear dimensionality reduction (Belkin & Partha, 2002) to clustering (Shi & Malik, 2000; Ng et al., 2002) to hashing (Weiss et al., 2009) to graph CNNs (Bruna et al., 2014) to spherical CNNs (Cohen et al., 2018) and to inference networks (Pfau et al., 2019). Spectral methods have been applied to learning tasks on spheres (Cohen et al., 2018) and graphs (Bruna et al., 2014), but not yet on hyperbolic spaces. This paper studies a spectral generalization of the FC (affine) layer on hyperbolic spaces.

Before presenting the spectral generalization of the affine layer, we introduce some notations. Let $(\cdot, \cdot)_E$ be the inner product, $|\cdot|$ the Euclidean norm, and $\rho$ an activation function. The Poincaré ball model of the hyperbolic space $\mathbf{H}^n (n \geq 2)$ is a manifold $\{x \in \mathbf{R}^n : |x| < 1\}$ equipped with a Riemannian metric $ds_{\mathbf{H}^n}^2 = \sum_{i=1}^n 4(1-|x|^2)^{-2} dx_i^2$. The boundary of $\mathbf{H}^n$ under its canonical embedding in $\mathbf{R}^n$ is the unit sphere $S^{n-1}$. The classical neuron $y = \rho((x, w)_E + b)$ is of input $x \in \mathbf{R}^n$, output $y \in \mathbf{R}$, with trainable parameters $w \in \mathbf{R}^n, b \in \mathbf{R}$. An affine layer $\mathbf{R}^n \to \mathbf{R}^m$ is a concatenation of $m$ neurons. An alternative representation of the neuron $x \mapsto \rho((x, w)_E + b)$ is given by [1]

$$x \in \mathbf{R}^n \mapsto \rho(\lambda(x, \omega)_E + b), \quad \omega \in S^{n-1}, \lambda, b \in \mathbf{R}. \tag{1}$$

This neuron is constant over any hyperplane that is perpendicular to a fixed direction $\omega$. In $\mathbf{H}^n$, a horocycle is a $n-1$ dimensional sphere (one point deleted) that is tangential to $S^{n-1}$. Horocycles are hyperbolic counterparts of hyperplanes (Bonola, 2012). Horocyclic waves $\langle x, \omega \rangle_H := \frac{1}{2} \log \frac{1-|x|^2}{|x-\omega|^2}$ are constant over any horocycle that is tangential to $S^{n-1}$ at $\omega$. Therefore,

$$x \in \mathbf{H}^n \mapsto \rho(\lambda \langle x, \omega \rangle_H + b), \quad \omega \in S^{n-1}, \lambda, b \in \mathbf{R} \tag{2}$$

---

[1] if $w \neq (0, \ldots, 0)$, one can take $\omega = w/|w|, \lambda = |w|$; else, one can take $\lambda = 0$ and any $\omega \in S^{n-1}$.

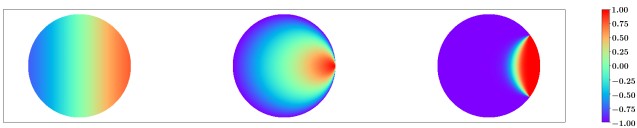

Figure 1: (Left) $\rho((\cdot,\omega)_E)$; (middle) $\rho(\langle\cdot,\omega\rangle_H)$; (right) $\rho(f^1_{a,p}(\cdot))$. In this figure, $\omega=(1,0)$, $a=(1,0), p=(0.5,0)$, and $\rho$ is tanh. The colorbar represents function values.

generalizes the classical neuron model (1), and a concatenation of finitely many (2) generalizes the FC (affine) layer. We call (2) a horocycle neuron. Figure 1 (middle) is an example on $\mathbf{H}^2$.

The neuron models in (1, 2) are related to spectral theory because $(\cdot,\omega)_E$ (respectively $\langle\cdot,\omega\rangle_H$) are building blocks of the Euclidean (respectively hyperbolic) Laplacian eigenspace. Moreover, many $L^2$ spaces have a basis given by Laplacian eigenfunctions (Einsiedler & Ward, 2017). On one side, all Euclidean (respectively hyperbolic) eigenfunctions are some kind of "superposition" of $(\cdot,\omega)_E$ (respectively $\langle\cdot,\omega\rangle_H$). On the other side, neural networks based on (1) (respectively (2)) represent functions that are another kind of "superposition" of $(\cdot,\omega)_E$ (respectively $\langle\cdot,\omega\rangle_H$). They heuristically explain why the universal approximation property is likely to hold for networks constructed by (1) and (2). By using the Hahn Banach theorem, an injectivity theorem of Helgason, and integral formula, we prove that finite sums of horocycle neurons (2) are universal approximators (Theorem 2).

Let $p \in \mathbf{H}^n$, $T_p(\mathbf{H}^n)$ be the tangent space of $\mathbf{H}^n$ at $p$, $a \in T_p(\mathbf{H}^n)$, $\oplus$ be the Möbius addition (Ungar, 2008). We remind the reader that the following functions

$$f^1_{a,p}(x) = \frac{2|a|}{1-|p|^2} \sinh^{-1} \left( \frac{2(-p \oplus x, a)_E}{(1-|-p \oplus x|^2)|a|} \right) \tag{3}$$

are building blocks of many hyperbolic learning tools (Ganea et al., 2018a; Mathieu et al., 2019; Shimizu et al., 2020). Figure 1 illustrates examples of different neuron models (1, 2, 3) on $\mathbf{H}^2$.

In Lemma 1, we shall present a close relationship between (2) and (3). Using this relationship and Theorem 2, we obtain a novel result on the expressivity of $f^1_{a,p}$ (Corollary 1).

This article contributes to hyperbolic learning. We first apply spectral methods, such as the horocycle, to hyperbolic deep learning. We prove results on the expressivity of horocycle neurons (2) and $f^1_{a,p}$ (3). With horocycle neurons, we obtain state-of-the-art results on the Poincaré-embedding subtree classification task and the classification accuracy of the 2-D visualization of images in in the experiment.

## 2 RELATED WORK

**Universal approximation** There is a vast literature on universal approximation (Cybenko, 1989; Hornik et al., 1989; Funahashi, 1989; Leshno et al., 1993). Cybenko (1989)'s existential approach uses the Hahn Banach theorem and Fourier transform of Radon measures. To prove Theorem 2, we also use the Hahn Banach theorem, and additionally an integral formula (7) and an injectivity Theorem 1 of Helgason. Generalizing integral formulas and injectivity theorems is easier than generalizing Fourier transform of Radon measures on most non-Euclidean spaces. (Carroll & Dickinson, 1989) uses the inverse Radon transform to prove universal approximation theorems. This method relates to ours, as injectivity theorems are akin to inverse Radon transforms. However, using the injectivity theorem is an existential approach while using the inverse Radon transform is a constructive one.

**Spectral methods** Spectral methods in Bronstein et al. (2017); Bruna et al. (2014); Cohen et al. (2018) use a basis of $L^2(X)$ given by eigenfunctions, where $X$ is a finite graph or the sphere. Because $L^2(\mathbf{H}^n)$ has no eigenfunctions as a basis, our approach is different from theirs.

**Hyperbolic deep learning** One part of hyperbolic learning concerns embedding data into the hyperbolic space (Nickel & Kiela, 2017; Sala et al., 2018). Another part concerns learning architectures with hyperbolic data as the input (Ganea et al. (2018a); Cho et al. (2019)). Ganea et al. (2018a) proposes two ways to generalize the affine layer on hyperbolic spaces: one by replacing the linear and bias part of an affine map with (25, 26) of their paper; another one by using a concatenation of $f^1_{a,p}$ in

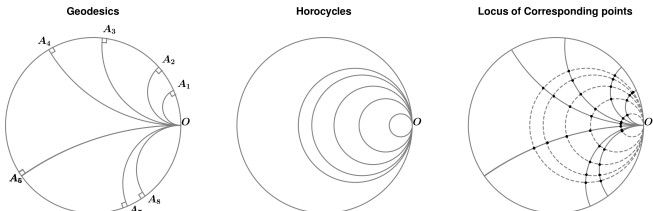

Figure 2: (Left) $A_i O$ are pairwise parallel lines in the sense of Gauss (Bonola, 2012); (Middle) A family of horocycles tangential to $O$; (Right) Each Horocycle is a locus of corresponding points (Bonola, 2012)[p.73] of parallel lines. This fact justifies that horocycles are analogs of hyperplanes.

their hyperbolic multiple linear regression (MLR). The latter seems more relevant to ours. A level set of $f_{a,p}^1$ is a hypercycle that has the same distance to a chosen geodesic hypersurface, while a level set of a horocycle neuron is a horocycle that has the same "spectral" distance to an ideal point at infinity. Based on functions similar to $f_{a,p}^1$, Mathieu et al. (2019); Shimizu et al. (2020) build the gyroplane layer and Poincaré FC layer. Ganea et al. (2018a); Cho et al. (2019) take geodesics as decision hyperplanes, while we (initially) take horocycles. We shall construct the horocycle multiple linear regression (MLR), where decision hypersurfaces are geodesics. Geodesics decision hyperplanes (Ganea et al., 2018a; Cho et al., 2019) and geodesic decision hypersurfaces here arise from different methods. Khrulkov et al. (2020) investigates hyperbolic image embedding, where prototypes (or models) of each class are center-based. We study a different one, and we shall call our prototypes end-based.

## 3 HYPERBOLIC SPACES

This section reviews facts from hyperbolic geometry that are used in the proof of Theorem 2. For the reader who is not interested in the proof, (4) is enough for the implementation.

**Hyperbolic metric**  We use the Poincaré model. The hyperbolic space $\mathbf{H}^n (n \geq 2)$ is the manifold $\{x \in \mathbf{R}^n : |x| < 1\}$ equipped with a Riemannian metric $ds^2 = \sum_{i=1}^n 4(1-|x|^2)^{-2} dx_i^2$. Let $o$ be the origin of $\mathbf{H}^n$. The distance function $d_{\mathbf{H}^n}$ satisfies $d_{\mathbf{H}^n}(o, x) = 2 \operatorname{arctanh} |x|$.

**Geodesics, horocycles and corresponding points**  Geodesics in $\mathbf{H}^n$ are precisely circular arcs that are orthogonal to $S^{n-1}$. Horocycles in $\mathbf{H}^n$ are precisely $(n-1)$-dimensional spheres that are tangential to $S^{n-1}$ (Helgason, 1970). Horocycles are hyperbolic analogs of hyperplanes. Figure 2 illustrates geodesics and horocycles on $\mathbf{H}^2$.

**Hyperbolic Poisson kernel**  The Poisson kernel for $\mathbf{H}^n$ is  $P(x, \omega) = \left( \frac{1-|x|^2}{|x-\omega|^2} \right)^{n-1}$ , where $x \in \mathbf{H}^n, \omega \in S^{n-1}$ (Helgason (1970)[p.108]). The function $\langle \cdot, \omega \rangle_H$ defined by

$$\langle x, \omega \rangle_H = \frac{1}{2(n-1)} \log P(\cdot, \omega) = \frac{1}{2} \log \frac{1-|x|^2}{|x-\omega|^2} \tag{4}$$

is constant over any horocycle that is tangential to $S^{n-1}$ at $\omega$ (Figure 1 (middle), (6)).

**Riemannian volume**  The Riemannian volume induced by the metric $ds^2$ on $\mathbf{H}^n$ is

$$dVol = 2^n (1 - |x|^2)^{-n} dx_1 \ldots dx_n. \tag{5}$$

**Horocycles**  Let $\Xi$ be the set of horocycles of $\mathbf{H}^n$, and let $\Xi_\omega$ be the set of all horocycles that are tangential to $S^{n-1}$ at $\omega$. Given $\lambda \in \mathbf{R}$, we let $\xi_{\lambda,\omega}$ be the unique horocycle that connects $\omega$ and $\tanh(\lambda/2) \cdot \omega$. We have $\Xi_\omega = \cup_{\lambda \in \mathbf{R}} \{\xi_{\lambda,\omega}\}$ and $\Xi = \cup_{\omega \in S^{n-1}} \Xi_\omega$. The length of any geodesic (that ends at $\omega$) line segment cut by $\xi_{\lambda_1,\omega}$ and $\xi_{\lambda_2,\omega}$ equals $|\lambda_1 - \lambda_2|$ (A.2). Therefore $|\lambda_1 - \lambda_2|$ is a natural distance function defined on $\Xi_\omega$, and the map $\lambda \to \xi_{\lambda,\omega}$ is an isometry between $\mathbf{R}$ and $\Xi_\omega$. This isometry is closely related to $\langle \cdot, \omega \rangle_H$ (A.3): for any $x \in \xi_{\lambda,\omega}$,

$$\langle x, \omega \rangle_H = \lambda/2. \tag{6}$$

The annoying $/2$ in (6) is a tradeoff that the metric here is different from that in Helgason (2000).

**Integral formula**    For fixed $\omega \in S^{n-1}$, $\mathbf{H}^n = \cup_{\lambda \in \mathbf{R}} \xi_{\lambda,\omega}$. Let $dVol_{\xi_{\lambda,\omega}}$ be the measure induced by $ds^2$ on $\xi_{\lambda,\omega}$. Let $L$ be a family of geodesics that end at $\omega$, $\delta > 0$, and $U = L \cap (\cup_{\lambda \le \alpha \le \lambda+\delta} \xi_{\alpha,\omega})$. For $l \in L$, $d_{\mathbf{H}}(l \cap \xi_{\lambda,\omega}, l \cap \xi_{\lambda+\delta,\omega}) = \delta$ (A.2), hence $dVol(U) = \delta \cdot dVol_{\xi_{\lambda,\omega}}(U \cap \xi_{\lambda,\omega})$ and therefore

$$\int_{\mathbf{H}^n} f(x) dVol(x) = \int_{\mathbf{R}} \left( \int_{\xi_{\lambda,\omega}} f(z) dVol_{\xi_{\lambda,\omega}}(z) \right) d\lambda. \tag{7}$$

The above proof (for $\mathbf{H}^n$) is essentially the same as that in (Helgason, 2000)[p.37] (for $\mathbf{H}^2$). To further convince the reader that (7) holds for all $n$, we give another simple proof in A.4.

**Injectivity theorem**    With respect to the canonical measure on $\Xi$, Helgason (1970)[p.13] proved
**Theorem 1** (Helgason). *If $f \in L^1(\mathbf{H}^n)$ and $\int_\xi f(z) dVol_\xi(z) = 0$ for a.e $\xi \in \Xi$, then $f = 0$ a.e..*

Theorem 1 demonstrates that if the integral of $f \in L^1(\mathbf{H}^n)$ over almost every horocycle is zero then $f$ is also zero. This theorem and the integral formula (7) are essential for the proof of Theorem 2.

# 4 LEARNING ARCHITECTURES AND EIGENFUNCTIONS OF THE LAPLACIAN

In this section, we discuss a heuristic connection between the representation properties of eigenfunctions and classical neurons, and then we define some horocycle-related learning tools.

## 4.1 EIGENSPACES AND NEURON MODELS

On a Riemannian manifold $X$, the Laplace-Beltrami $L_X$ is the divergence of the gradient, and it has a well-known representation property (Einsiedler & Ward, 2017): if $X$ is a compact Riemannian manifold or bounded domain in $\mathbf{R}^n$, then $L^2(X)$ has a basis given by eigenfunctions. This statement is false if $X$ is $\mathbf{R}^n$ or $\mathbf{H}^n$ (Hislop, 1994).

**Eigenspaces of on $\mathbf{R}^n$ and $\mathbf{H}^n$**    Our work is motivated by the theory of eigenspaces, in which Euclidean (respectively hyperbolic) eigenfunctions are obtained from $(x,\omega)_E$ (respectively $\langle x,\omega \rangle_H$) by some kind of superposition. For example, all smooth eigenfunctions of $L_{\mathbf{R}^n}$ are precisely the functions (M. Hashizume & Okamoto, 1972)[p.543]

$$f(x) = \int_{S^{n-1}} e^{\lambda(x,\omega)_E} dT(\omega), \tag{8}$$

and eigenfunctions of $L_{\mathbf{H}^n}$ are precisely the functions (Helgason, 1970)[Theorem 1.7, p.139]

$$f(x) = \int_{S^{n-1}} e^{\lambda\langle x,\omega \rangle_H} dT(\omega), \tag{9}$$

where $T$ in (8) and (9) are some technical linear forms of suitable functional spaces on $S^{n-1}$.

**Neuron models**    By (8) and (1), Euclidean eigenfunctions (respectively classical neurons) are superpositions of $(\cdot,\omega)_E$ and $\exp$ (respectively $\rho$), with homogeneity and additivity. By (9) and (2), hyperbolic eigenfunctions (respectively horocycle neurons) are superpositions of $\langle \cdot,\omega \rangle_H$ and $\exp$ (respectively $\rho$). The representation property of eigenfunctions on compact manifolds and bounded domains suggests that the universal approximation property is likely to hold for networks constructed by $(\cdot,\omega)_E$ or $\langle \cdot,\omega \rangle_H$. However, this heuristic is not proof (A.5).

## 4.2 HOROCYCLE BASED LEARNING ARCHITECTURES

**Horocycle neuron**    In the implementation of the horocycle neuron (2), we take $\frac{1}{2} \log \left( \frac{1-|x|^2}{|x-\omega|^2+\epsilon} + \epsilon \right)$ for $\langle x,\omega \rangle_H$, where $\epsilon$ is a small constant to ensure numerical stability. For updating $\omega$, we use the sphere optimization algorithm (Absil et al., 2008; Bonnabel, 2013) (A.6).

**Horocycle feature and horocycle decision hypersurface**    Given a non-origin point $x \in \mathbf{H}^n$, for $y \in \mathbf{H}^n$ we define $h_x(y) = \langle y, x/|x| \rangle_H$ and call it the horocycle feature attached to $x$. This feature is useful in the Poincaré embedding subtree classification task (see the experiment and Figure 3[left]). The horocycle is the hyperbolic analog of the Euclidean hyperplane, and therefore it could be a possible choice of decision hypersurface, which may arise from a level set of a horocycle feature.

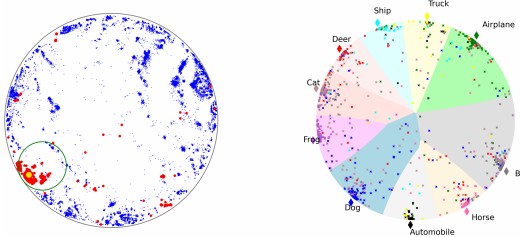

Figure 3: (Left) Horocycle decision hypersurface. Colored points form a Poincaré embedding of WordNet nouns in $\mathbf{H}^2$. The yellow point is group.n.01. Points from the subtree rooted at group.n.01 are in red, and the rest are in blue. The green horocycle separates the blue and the red. (Right) End-based clustering and geodesic decision hypersurfaces. It is an embedding of CIFAR-10 in $\mathbf{H}^2$. Different classes are in different colors. Thin diamonds on $S^1$ are prototypes. Decision regions are separated by geodesic decision hypersurfaces.

**End-based clustering and end prototype**   Natural clustering is a topic in representation learning (Bengio et al., 2013), and the common prototype-based clusters are center-based (Tan et al., 2005). We propose a type of clustering that embeds high-dimensional data in $\mathbf{H}^n$ and places prototypes in $S^{n-1}$. Figure 3[right] is an example for $n = 2$. For $\omega \in S^{n-1}$ and any $b \in \mathbf{R}$, the function $x \in \mathbf{H}^n \mapsto -\log\left(\frac{1-|x|^2}{|x-\omega|^2}\right) + b$ measures the relative distance of $\mathbf{H}^n$ from $\omega$ in Gromov's bordification theory (Bridson & Haefliger (2009)[II.8], A.18). Moreover, we define $\mathrm{Dist} : \mathbf{H}^n \times S^{n-1} \times \mathbf{R} \to \mathbf{R}$ by

$$\mathrm{Dist}(x, \omega, b) = -\log\left(\frac{1-|x|^2}{|x-\omega|^2}\right) + b = -2\langle x, \omega \rangle_H + b. \tag{10}$$

It is a relative distance function, and this is why $\mathrm{Dist}$ may assume negative values and why there is a bias term $b$ in (10). Consider classes $\mathrm{Cls} = \{C_1, C_2, \ldots, C_M\}$ and labeled training examples $\{(X^1, Y^1), \ldots, (X^N, Y^N)\}$, where $X^i \in \mathbf{R}^D$ are $D$-dimensional input features and $Y^i \in \{1, 2, \ldots, M\}$. Each example $X^i$ belongs to the class $C_{Y^i}$. In light of (10), our goal is to find a neural network $\mathrm{NN}_\theta : \mathbf{R}^D \to \mathbf{H}^n$ that is parameterized by $\theta$, prototypes $\omega_1, \ldots, \omega_M \in S^{n-1}$, and real numbers $b_1, \ldots, b_M \in \mathbf{R}$ such that

$$\frac{\#\left\{1 \leq i \leq N : Y^i = \underset{1 \leq j \leq M}{\arg\min}\left(\mathrm{Dist}(\mathrm{NN}_\theta(X^i), \omega_j, b_j)\right)\right\}}{N} \tag{11}$$

is maximized. We call $\{\mathrm{NN}_\theta(X^j) : 1 \leq j \leq N\}$ the end-based clustering and $\omega_i$ end prototypes (in hyperbolic geometry, the end is an equivalence class of parallel lines in Figure 2[left]). In experiments, we take $\mathrm{NN}_\theta = \mathrm{Exp} \circ \mathrm{NN}'_\theta$, where $\mathrm{NN}'_\theta : \mathbf{R}^D \to \mathbf{R}^n$ is a standard neural network parameterized by $\theta$ and $\mathrm{Exp} : \mathbf{R}^n \to \mathbf{H}^n$ is the exponential map of the hyperbolic space.

**Horocycle layer, horocycle multiple linear regression (MLR) and geodesic decision hypersurfaces**   We call a concatenation of (2) a horocycle layer, and we shall carefully describe a prototypical learning framework for end-based clusterings. Using the same notions as in the previous paragraph, the classification task has $M$ classes, and $\mathrm{NN}_\theta = \mathrm{Exp} \circ \mathrm{NN}'_\theta : \mathbf{R}^D \to \mathbf{H}^n$ is a deep network. For prototypes $\omega_1, \ldots, \omega_M \in S^{n-1}$, real numbers $b_1, \ldots, b_M \in \mathbf{R}$, and any example $X$, our feedforward for prediction will be

$$x = \mathrm{NN}_\theta(X), \qquad \text{(Feature descriptor)}$$
$$\mathrm{SC}_j(X) = -\mathrm{Dist}(x, \omega_j, b_j), \qquad \text{(Scores; Similarity)}$$
$$X \in C_{\underset{1 \leq j \leq M}{\arg\max(\mathrm{SC}_j(X))}}. \qquad \text{(Classifier)}$$

The goal is to maximize the accuracy (11), and then we need a loss function for the backpropagation. Following the convention of prototypical networks (Snell et al., 2017; Yang et al., 2018), we choose an increasing function $\rho$ (in our experiments, $\rho(x) = x$ or $\rho = \tanh$. [2]) and let the distribution over classes for an input $X$ (with label $Y$) be

$$p_\theta(Y = C_j | X) \propto e^{-\rho(\mathrm{Dist}(\mathrm{NN}_\theta(X), m_j, b_j))} = e^{-\rho(-\mathrm{SC}_j(X))}.$$

---

[2]One often takes $\rho(x) = x^2$ in metric learning, which is improper here because $\mathrm{Dist}(x)$ could be negative.

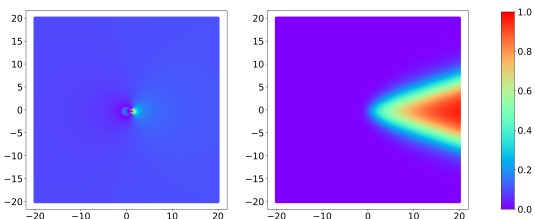

Figure 4: Prediction probabilities of classifiers. Suppose the classification task has 10 classes. Let $\text{NN}_\theta : \mathbf{R}^D \to \mathbf{R}^2$ be the feature descriptor, $X$ the input, and $x = \text{NN}_\theta(X)$. Let $w_i = (\cos((i-1)\pi/5), \sin((i-1)\pi/5))(1 \le i \le 10)$. (Left) $p_\theta(Y = C_1|x)$ on $\mathbf{R}^2$ when score functions are Poisson neurons $\text{SC}_j(X) = \text{BatchNorm}(\text{P}^\rho_{w_j,-1,0}(x))$. (Right) $p_\theta(Y = C_1|x)$ on $\mathbf{R}^2$ when score functions are $\text{SC}_j(X) = (x, w_j)_E$.

Therefore, given a batch of training examples, the loss function is

$$L = \frac{-\sum_{(X^j, Y^j) \in \text{Batch}} \log p_\theta(Y = C_{Y^j}|X_j)}{\#\text{Batch}}. \tag{12}$$

The training proceeds by minimizing $L$, and we call this framework a horocycle MLR. The set of parameters of the framework is $\{\theta\} \cup \{\omega_1, \ldots, \omega_M\} \cup \{b_1, \ldots, b_M\}$. It is worth mentioning that decision boundaries of the horocycle MLR are geodesics, which follows from

$$\text{SC}_i(X) = \text{SC}_j(X) \iff \log\left(\frac{1-|x|^2}{|x-\omega_i|^2}\right) - b_i = \log\left(\frac{1-|x|^2}{|x-\omega_j|^2}\right) - b_j \iff \frac{|x-\omega_i|}{|x-\omega_j|} = e^{\frac{b_j - b_i}{2}}$$

and the theorem of Apollonian circles (A.7).

**Poisson neuron and Poisson multiple linear regression (MLR)** Although $\langle x, \omega \rangle_H$ (4) is well-motivated by the theory of eigenspaces (9) and fits naturally into metric learning (see 10 or also Corollary 1), it is only defined on $\mathbf{H}^n$. Some readers might not be convinced that the neuron has to be defined on hyperbolic spaces. Therefore, we try to remove the $\log$ in (4) and define the Poisson neuron model by $\text{P}^\rho_{w,\lambda,b}(x) = \rho\left(\lambda \frac{|w|^2 - |x|^2}{|x-w|^2} + b\right)$ for $w \in \mathbf{R}^n, \lambda, b \in \mathbf{R}$, which is well-defined on $\mathbf{R}^n \backslash \{w\}$. Notice that if $|x| < |w|$ then $\frac{|w|^2 - |x|^2}{|x-w|^2} = e^{2\langle x/|w|, w/|w| \rangle_H}$. In A.8, Figure 7 illustrates an example of a Poisson neuron on $\mathbf{R}^2$. In the implementation, we take $\frac{|w|^2 - |x|^2}{|x-w|^2 + \epsilon}$ for $\frac{|w|^2 - |x|^2}{|x-w|^2}$, where $\epsilon$ is a small constant for numerical stability. We call a concatenation of Poisson neurons a Poisson layer, and we use it with a deep neural network $\text{NN}_\theta : \mathbf{R}^D \to \mathbf{R}^n$ to construct the Poisson MLR, which is similar to the horocycle MLR. Let $w_1, \ldots, w_M \in \mathbf{R}^n$ and $b_1, \ldots, b_M \in \mathbf{R}$, the feedforward for prediction of our framework is

$$x = \text{NN}_\theta(X), \text{SC}_j(X) = \text{BatchNorm}(\text{P}^\rho_{w_j,-1,b_j}(x)), X \in C_{\underset{1 \le j \le M}{\arg\max}(\text{SC}_j(X))}. \tag{13}$$

We let the $p_\theta(Y = C_j|X) \propto e^{\text{SC}_j(X)}$ and take (12) as the loss. This framework is called a Poisson MLR. We use the usual optimization algorithms to update parameters in the Poisson neuron. The BatchNorm(Ioffe & Szegedy, 2015) seems crucial for (13) in the experiment. Figure 4 illustrates that high-confidence prediction regions (deep red areas) of the Poisson MLR are compact sets, in contrast to classical classifiers Hein et al. (2019)[Theorem 3.1]. We shall use this figure to explain an experiment in Section 6.4.

## 5 REPRESENTATIONAL POWER

In this section, $\rho$ is a continuous sigmoidal function (Cybenko, 1989), ReLU(Nair & Hinton, 2010), ELU(Clevert et al., 2016), or Softplus(Dugas et al., 2001). We remind the reader that $\rho$ is sigmoidal if $\lim_{t \to \infty} \rho(t) = 1$ and $\lim_{t \to -\infty} \rho(t) = 0$. The following theorem justifies the representational power of horocycle neurons.

**Theorem 2.** *Let $K$ be a compact set in $\mathbf{H}^n$, and $1 \le p < \infty$. Then finite sums of the form*

$$F(x) = \sum_{i=1}^N \alpha_i \rho(\lambda_i \langle x, \omega_i \rangle_H + b_i), \quad \omega_i \in S^{n-1}, \alpha_i, \lambda_i, b_i \in \mathbf{R} \tag{14}$$

*are dense in $L^p(K, \mu)$, where $\mu$ is either $d\text{Vol}$ (5) or the induced Euclidean volume.*

We provide a sketch of the proof here and go through the details in A.9. It suffices to prove the theorem for a sigmoidal function $\rho$ and $\mu = d\,Vol$, as other cases follow from this one. Assume that these finite sums are not dense in $L^p(K, d\,Vol)$. By the Hahn-Banach theorem, there exists some nonzero $h \in L^q(K, d\,Vol)$, where $q = p/(p-1)$ if $p > 1$ and $q = \infty$ if $p = 1$, such that $\int_K F(x)h(x)d\,Vol(x) = 0$ for all finite sums of the form (14). Extend $h$ to be a function $H$ that is defined on $\mathbf{H}^n$ by assigning $H(x) = h(x)$ if $x \in K$ and $H(x) = 0$ if $x \in \mathbf{H}^n \backslash K$. Using the property of sigmoidal functions, the bounded convergence theorem, and the integral formula (7), we prove that the integration of $H$ on almost every horocycle is zero. By the injectivity Theorem 1, $H$ is almost everywhere zero, which contradicts our assumption and completes the proof.

In A.10, we shall prove the same result for Poisson neurons. In A.11, we prove the following lemma, which demonstrates a close relationship between horocycle neurons and the widely used $f_{a,p}^1$ (3).

**Lemma 1.** *Let $K$ be a compact set in $\mathbf{H}^n$, $\omega \in S^{n-1}$, and $\epsilon > 0$. There are $c, d \in \mathbf{R}$, $p \in \mathbf{H}^n$, and $a \in T_p(\mathbf{H}^n)$ such that the function $D(x) = c f_{a,p}^1(x) + d - \langle x, \omega \rangle_H$ satisfies $||D||_{L^p(K, d\,Vol)} < \epsilon$.*

This lemma suggests that $\langle \cdot, \omega \rangle_H$ is a boundary point of some "compactification" of the space of $f_{a,p}^1$. The above lemma together with Theorem 2 implies

**Corollary 1.** *Let $K$ be a compact set in $\mathbf{H}^n$ and $1 \leq p < \infty$. Finite sums of the form*

$$F(x) = \sum_{i=1}^{N} \alpha_i \rho(c_i f_{a_i, p_i}^1(x) + d_i), \ \ p_i \in \mathbf{H}^n, a_i \in T_{p_i}(\mathbf{H}^n), \alpha_i, c_i, d_i \in \mathbf{R},$$

*are dense in $L^p(K, \mu)$, where $\mu = d\,Vol$ or $\mu$ is the induced Euclidean volume.*

This result provides novel insights into the hyperbolic neural network (Ganea et al., 2018a), gyroplane layer (Mathieu et al., 2019), and Poincaré FC layer (Shimizu et al., 2020). Although level sets of $f_{a,p}^1$ are hypercycles, our proof of Lemma 1 relies on the theory of horocycles. It would be interesting to have more natural approaches to treat the expressivity of $f_{a,p}^1$.

## 6 EXPERIMENTS

In this section, we first play with the MNIST toy. Next, we apply a horocycle feature to the Poincaré embedding subtree classification task. After that, we construct 2-D clusterings of image datasets by using the horocycle MLR. Finally, we provide evidence for further possible applications of the Poisson MLR. We use the framework or some functions of Tensorflow, Keras, and scikit-learn (Abadi et al., 2015; Chollet et al., 2015; Pedregosa et al., 2011).

### 6.1 MNIST

The MNIST (LeCun et al., 1998) task is popular for testing hyperbolic learning tools (Ontrup & Ritter, 2005; Nagano et al., 2019; Mathieu et al., 2019; Grattarola et al., 2019; Ovinnikov, 2020; Khrulkov et al., 2020). We train two different classifiers. A.12, A.14, and code contain details. The first one is a single horocycle layer followed by the softmax classifier. The average test error rate after 600 epochs is **1.96**%, and Theorem 2 provides the rationale for this experiment (A.13). The second one is a Poisson MLR. It is the best hyperbolic geometry related MNIST classifier (Table 1). In this table, Ontrup & Ritter (2005) uses the hyperbolic SOM, Grattarola et al. (2019) uses the adversarial autoencoder, and Khrulkov et al. (2020) uses the hyperbolic MLR. Our experiment performs well on MNIST suggests that horocycle and Poisson neurons are computationally efficient and easily coordinate with classical learning tools (such as the convolutional layer and the softmax).

Table 1: Test error rates of hyperbolic geometry related MNIST classifiers

| Ontrup & Ritter (2005) | Grattarola et al. (2019) | Khrulkov et al. (2020) | This paper |
|---|---|---|---|
| 5.4% | 4.2% | 1% | **0.35**% |

## 6.2 POINCARÉ EMBEDDING SUBTREE CLASSIFICATION

Given a Poincaré embedding (Nickel & Kiela, 2017) $\mathcal{PE} : \{\text{WordNet noun}\} \to \mathbf{H}^D$ of 82114 nouns and given a node $x \in \{\text{WordNet noun}\}$, the task is to classify all other nodes as being part of the subtree rooted at $x$ (Ganea et al., 2018a). Our model is logistic regression, where the horocycle feature $p \in \{\text{WordNet noun}\} \mapsto h_{\mathcal{PE}(x)}(\mathcal{PE}(p)/s)$ ($s$ is a hyperparameter lying in $[1, 1.5]$) is the only predictor, and the dependent variable is whether $p$ is in the subtree rooted at $x$. The decision hypersurface of this model is a horocycle, as illustrated in Figure 3 (left). In the experiment, we pre-train three different Poincaré embeddings[3] in each of $\mathbf{H}^2, \mathbf{H}^3, \mathbf{H}^5, \mathbf{H}^{10}$. For each $x \in \{\text{animal, group, location, mammal, worker}\}$ and $D \in \{2, 3, 5, 10\}$, we randomly select one of three pre-trained Poincaré embedding $\mathcal{PE} : \{\text{WordNet noun}\} \to \mathbf{H}^D$ and then test the model.

Table 2 reports the F1 classification scores and two standard deviations of 100 trials for each $\{x, D\}$. Different Poincaré embeddings account for the most variance of the performance. Our model is different from the existing ones. Firstly, we take the horocycle as the decision hypersurface, while others take the geodesic. Secondly, we train a logistic regression on top of the horocycle feature attached to $\mathcal{PE}(x)$, which is efficiently calculated, while others train the hyperbolic MLR with different parametrizations. On the number of parameters, we have three (independent of $D$), Ganea et al. (2018a) has $2D$, and Shimizu et al. (2020) has $D + 1$. The number of parameters explains why our model is prominent in low dimensions.

Table 2: Average test F1 classification scores (%) for five subtrees of WordNet noun tree

| RootNode | Model | $\mathbf{H}^2$ | $\mathbf{H}^3$ | $\mathbf{H}^5$ | $\mathbf{H}^{10}$ |
|---|---|---|---|---|---|
| animal.n.01 | Ganea et al. (2018a) | 47.43±1.07 | **91.92±0.61** | 98.07±0.55 | **99.26±0.59** |
| 4016 nodes | Shimizu et al. (2020) | 60.69±4.05 | 67.88±1.18 | 86.26±4.66 | 99.15±0.46 |
|  | This paper | **94.32±5.33** | 86.98±2.05 | **98.57±4.98** | 98.76±2.43 |
| group.n.01 | Ganea et al. (2018a) | 81.72±0.17 | 89.87±2.73 | 87.89±0.80 | 91.91±3.07 |
| 8376 nodes | Shimizu et al. (2020) | 74.27±1.50 | 63.90±6.46 | 84.36±1.79 | 85.60±2.75 |
|  | This paper | **90.08±12.41** | **90.91±11.62** | **97.52±1.15** | **97.85±1.92** |
| location.n.01 | Shimizu et al. (2020) | 42.60±2.69 | 66.70±2.67 | 78.18±5.96 | 92.34±1.84 |
| 3362 nodes | This paper | **93.19±11.76** | **94.66±4.65** | **95.23±4.11** | **97.37±1.75** |
| mamal.n.01 | Ganea et al. (2018a) | 32.01±17.14 | 87.54±4.55 | 87.73±3.22 | 91.37±6.09 |
| 1181 nodes | Shimizu et al. (2020) | 63.48±3.76 | 94.98±3.87 | **99.30±0.30** | **99.17±1.55** |
|  | This paper | **98.74±1.93** | **96.37±2.42** | 93.34±8.86 | 95.80±1.62 |
| worker.n.01 | Ganea et al. (2018a) | 12.68±0.82 | 24.09±1.49 | 55.46±5.49 | 66.83±11.38 |
| 1115 nodes | This paper | **94.47±4.30** | **92.53±6.36** | **95.47±2.71** | **94.75±2.18** |

## 6.3 END-BASED CLUSTERING FOR 2D DIMENSION REDUCTION

In this experiment, we use the horocycle MLR (Section 4.2) to construct end-based clusterings $\text{NN}_\theta : \mathbf{R}^D \to \mathbf{H}^2$ for MNIST, Fashion-MNIST(Xiao et al., 2017), and CIFAR-10(Krizhevsky, 2012). We take $\text{NN}_\theta = \text{Exp} \circ \text{NN}'_\theta$, where $\text{Exp}$ is the exponential map of $\mathbf{H}^2$ and $\text{NN}'_\theta : \mathbf{R}^D \to \mathbf{R}^2$ is a network with four convolutional blocks for MNIST/Fashion-MNIST or a ResNet-32 structure for CIFAR-10. A.16 and code contain details.

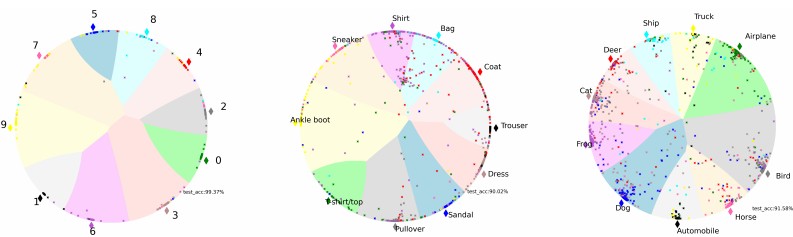

Figure 5: End-based clusters of MNIST, Fashion-MNIST, and CIFAR-10. Test error rates from the left to the right: 0.63%, 9.98%, 8.42%. The performance for CIFAR-10 is better than Fashion-MNIST because $\text{NN}'_\theta$ is of a ResNet-32 structure for CIFAR-10 but of only four convolutional layers for Fashion-MNIST. Thin diamonds are prototypes, and shallow areas are decision regions.

---

[3]https://github.com/dalab/hyperbolic_cones

Figure 5 illustrates end-based clusterings for MNIST, Fashion-MNIST, and CIFAR-10, with performance reported in the caption. Our accuracy for Fashion-MNIST is $8\%$ higher than all numbers presented in McInnes et al. (2020). Moreover, Table 3 compares the numbers of Yang et al. (2018); Ghosh & Kirby (2020), and ours for MNIST, and our methods are similar. We all use convolutional networks as the (Feature descriptor) and prototype-based functions as the loss. However, Yang et al. (2018); Ghosh & Kirby (2020) use the center-based prototype loss, while we use the end-based (12). Yang et al. (2018)[Figure 1] points out that the traditional CNN is good at linearly separating feature representations, but the learned features are of large intra-class variations. The horocycle MLR leads to the inter-class separability in the same way (angle accounts for label difference) a traditional CNN does. At the same time, it also obtains intra-class compactness (Figure 5).

Table 3: Test error rates on 2D embedded MNIST by dimensionality reduction techniques

| GerDA | Centroidencoder | dG-MCML | dt-MCML | CPL | GCPL | This paper |
|-------|-----------------|---------|---------|------|------|------------|
| 3.2% | 2.61% | 2.13% | 2.03% | 0.72% | 0.67% | **0.63**% |

### 6.4 POISSON MLR

Using a Poisson MLR whose feature descriptor is a ResNet-32 structure, we obtain a classifier with a test error rate of $\mathbf{6.46}\%$ on CIFAR-10. It is on par with other methods with similar network structures (Yang et al., 2018). Moreover, we apply Poisson MLR to the classification task of flowers (Tensorflow), which is a typical example of overfitting. Replacing the MLR part of the Keras model (Tensorflow) with a Poisson MLR, the new Poisson model shows better generalization performance (Figure 6). A.17 and code contain the details. This subsection provides evidence for further applications of horocycles.

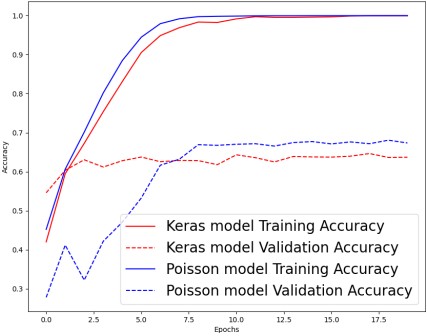

Figure 6: Keras'MLR and our Poisson MLR on the task of flowers. Figure 4 explains this experiment. In the earlier epochs of the training, feature vectors $\mathrm{NN}_\theta(X)$ of the Poisson model are not even close to its compact high-confidence prediction regions (deep red areas), and therefore on the test data, it is not much better than a random guess. In the end, feature vectors lying in these compact regions are of small intra-class variations, which is good for generalization (Yang et al., 2018).

## 7 CONCLUSION

Based on the spectral theory of hyperbolic spaces, we introduce several horocycle-related learning tools. They find applications in the hyperbolic neural networks, the Poincaré embedding subtree classification task, and the visualization and classification of image datasets. We give an existential proof of a universal approximation theorem for shallow networks constructed by horocycle neurons or $f_{a,p}^1$. Hopefully, it will trigger further research on the expressivity problems, such as constructive approaches, quantitative results, and benefit of depth (Mhaskar & Poggio, 2016), on horocycle neurons, $f_{a,p}^1$, and similar functions on more general manifolds.

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

# A APPENDIX

## A.1 NOTATIONS AND SYMBOLS

### Default Notations

| Notation | Description | Related formula |
|---|---|---|
| $\mathbf{R}$ | The set of real numbers | |
| $\mathbf{R}^n$ | $n$ dimensional Euclidean space | $x \in \mathbf{R}^n, x = (x_1, \ldots, x_n)$ |
| $(\cdot, \cdot)_E$ | Euclidean inner product | $x \in \mathbf{R}^n, y \in \mathbf{R}^n, (x, y)_E = \sum_{i=1}^n x_i y_i$ |
| $\langle \cdot, \cdot \rangle_H$ | Hyperbolic analogue of $(\cdot, \cdot)_E$ | $x \in \mathbf{H}^n, y \in S^{n-1}, \langle x, \omega \rangle_H = \frac{1}{2} \log \frac{1-|x|^2}{|x-\omega|^2}$ |
| $|\cdot|$ | Euclidean norm | $x \in \mathbf{R}^n, |x| = \sqrt{(x,x)_E}$ |
| $\mathbf{H}^n$ | $n$ dimensional hyperbolic space | as a set, $\mathbf{H}^n = \{x \in \mathbf{R}^n : |x| < 1\}$ |
| $T_p(X)$ | Tangent space of $X$ at $p$ | |
| $T(X)$ | Tangent space of $X$ | $T(X) = \cup_{p \in X} T_p(X)$ |
| $ds^2_{\mathbf{H}^n}$ | The canonical metric on $\mathbf{H}^n$ with curvature -1 | $ds^2_{\mathbf{H}^n} = \sum_{i=1}^n 4(1-|x|^2)^{-2} dx_i^2$ |
| $dVol$ | Riemannian volume on $\mathbf{H}^n$ | $dVol = 2^n(1-|x|^2)^{-n} dx_1 \ldots dx_n$ |
| $L^p(K, dVol)$ | $L^p$ space | $L^p(K, dVol) = \{f| \int_K |f|^p dVol < \infty\}$ |
| $||\cdot||_{L^p(K,dVol)}$ | $L^p$ norm | $f$ measurable on $K$, $||f||_{L^p(K,dVol)} = \left(\int_K |f|^p dVol\right)^{\frac{1}{p}}$ |
| $S^{n-1}$ | $n-1$ dimensional sphere | as a set, $S^{n-1} = \{x \in \mathbf{R}^n : |x| = 1\}$ |
| $P(\cdot, \cdot)$ | Hyperbolic Poisson kernel | $x \in \mathbf{H}^n, \omega \in S^{n-1}, P(x, \omega) = \left(\frac{1-|x|^2}{|x-\omega|^2}\right)^{n-1}$ |
| $f^1_{a,p}$ | Model in the hyperbolic MLR | $f^1_{a,p}(x) = \frac{2|a|}{1-|p|^2} \sinh^{-1}\left(\frac{2(-p \oplus x, a)_E}{(1-|-p \oplus x|^2)|a|}\right)$ |
| $d_{\mathbf{H}^n}$ | The hyperbolic distance function | |
| $\Xi$ | The space of horocycles | |
| $\Xi_\omega$ | The set of horocycles that are tangential to $S^{n-1}$ at $\omega$ | |
| $L_X$ | Laplace-Beltrami operator on $X$ | |
| $h_x$ | The horocycle feature function | $h_x(y) = \langle y, x/|x| \rangle_H$ |
| $\xi_{\lambda,\omega}$ | The unique horocycle connecting $\omega$ and $\tanh \lambda/2 \cdot \omega$. | |
| MLR | Multiple linear regression | |
| dim | dimension | |
| $I_K$ | the indicator function of $K$ | |
| Dist | Relative distance function | $\text{Dist}(x, \omega, b) = -2\langle x, \omega \rangle_H + b$ |
| Cls | Set of classes | $\text{Cls} = \{C_1, C_2, \ldots, C_M\}$ |
| $\text{NN}_\theta$ | A network parameterized by $\theta$ | |
| $\text{NN}'_\theta$ | A network parameterized by $\theta$ | |
| Exp | Exponential map of the hyperbolic space | |
| $(X^1, Y^1)$ | Labeled sample | |
| $\text{SC}_j$ | Score function | |
| $p_\theta(Y = C_j|X)$ | Prediction probability | |
| $L$ | Loss function | |
| $\text{P}^\rho_{w,\lambda,b}$ | Poisson neuron | $\text{P}^\rho_{w,\lambda,b}(x) = \rho\left(\lambda \frac{|w|^2-|x|^2}{|x-w|^2} + b\right)$ |
| $\mathcal{PE}$ | Poincaré embedding | |

### Conventional symbols

| Symbol | In most cases it refers |
|---|---|
| $n, m, i$ | integers |
| $x, y, w$ | points in $\mathbf{R}^n$ or $\mathbf{H}^n$, or real numbers |
| $o$ | the origin of $\mathbf{R}^n$ or $\mathbf{H}^n$ |
| $b, c, d, \alpha, \delta$ | real numbers |
| $\lambda$ | real or complex number |
| $t$ | real number, represent the timestamp in optimization |
| $\omega$ | point in $S^{n-1}$ |
| $\rho$ | an activation function |
| $f, g$ | functions |
| $K$ | a compact set |
| $X$ | a manifold |
| $p$ | a point in $\mathbf{H}^n$ or on a manifold |
| $a$ | an element in $T_p(\mathbf{H}^n)$ |
| $\xi$ | a horocycle |
| $\mu$ | a measure |
| $L$ | a family of geodesics lines |
| $l$ | a geodesics line |
| $U$ | a set in $\mathbf{H}^n$ |
| $F, h, H$ | functions |
| $M$ | number of classes |
| $D$ | dimension |

### A.2 PROOF OF THE ISOMETRY

Given $\omega \in S^{n-1}$ and $\lambda \in \mathbf{R}$, we let $\xi_{\lambda,\omega}$ the unique horocycle that connects $\omega$ and $\tanh(\lambda/2) \cdot \omega$. The length of any geodesic (that ends at $\omega$) line segment cut by $\xi_{\lambda_1,\omega}$ and $\xi_{\lambda_2,\omega}$ equals $|\lambda_1 - \lambda_2|$. This fact is obvious in the half-space model.

There is a Riemannian isometry $F : \{z \in \mathbf{R}^n : |z| < 1\} \to \{(x_1, \cdots, x_n) : x_1 > 0\}$ (the latter is with the metric $ds^2 = \frac{dx_1^2 + \cdots + dx_n^2}{x_1^2}$) such that $F(\omega) = \infty$ and $F(o) = (1, 0, \ldots, 0)$. Using $d_{\mathbf{H}^n}(o, \tanh(\lambda_i/2)\omega) = |\lambda_i|$, $d_{\{(x_1, \cdots, x_n) : x_1 > 0\}}((1, 0, \ldots, 0), (e^{\pm \lambda_i}, 0, \ldots, 0)) = |\lambda_i|$, $F(\omega) = \infty$ and $F(o) = (1, 0, \ldots, 0)$, we have $F(\tanh(\lambda_i/2)\omega) = (e^{\lambda_i}, 0, \ldots, 0)$. Therefore, $F$ maps $\xi_{\lambda_i,\omega}$ to $\{(x_1, x_2, \ldots, x_n) : x_1 = e^{\lambda_i}\}$. Any geodesic (that ends at $\omega$) line segment cut by $\xi_{\lambda_1,\omega}$ and $\xi_{\lambda_2,\omega}$ is mapped by $F$ to $\{(t, \alpha_2, \ldots, \alpha_n) : (t - e^{\lambda_1})(t - e^{\lambda_2}) < 0\}$ for some fixed $\alpha_j$. It is easy to check the length of this segment with respect to $\frac{dx_1^2 + \cdots + dx_n^2}{x_1^2}$ (as $\alpha_i$ are constants, the metric reduces to $dx_1^2/x_1^2$ on this segment) is $|\lambda_1 - \lambda_2|$.

### A.3 PROOF OF (6)

Because $x$ is on $\xi_\lambda$ which is a sphere with center $\frac{1+\tanh \lambda/2}{2}\omega$ and radius $\frac{1-\tanh \lambda/2}{2}$, we have $\left| x - \frac{1+\tanh \lambda/2}{2}\omega \right|^2 = \left| \frac{1-\tanh \lambda/2}{2} \right|^2$, which leads to $|x|^2 - (1+\tanh \lambda/2)(x, \omega)_E + \tanh \lambda/2|\omega|^2 =$

0, and then $\frac{1+\tanh\lambda/2}{2}|x-\omega|^2 = \frac{1-\tanh\lambda/2}{2}(|\omega^2|-|x|^2)$, and finally $\langle x,\omega\rangle_H = \frac{1}{2}\log\frac{|\omega|^2-|x|^2}{|x-\omega|^2} = \frac{1}{2}\log\frac{1+\tanh\lambda/2}{1-\tanh\lambda/2} = \lambda/2$.

### A.4 ANOTHER PROOF OF THE INTEGRAL FORMULA (7)

We use $H^n$ for the upper half space model $\{(x_1,\cdots,x_n) : x_1 > 0\}$ with the Riemannian volume $\frac{dx_1\cdots dx_n}{x_1^n}$. Let $\omega = (\infty,0,\ldots,0)$ and $o$ be $(1,0,\ldots,0)$ as in (A.2), then $\xi_{\lambda,\omega} = \{(x_1,x_2,\ldots,x_n) : x_1 = e^\lambda\}$. The induced Riemannian metric on $\xi_{\lambda,\omega}$ (respectively volume $dVol_{\xi_{\lambda,\omega}}$) is $\frac{dx_2^2+\cdots+dx_n^2}{e^{2\lambda}}$ (respectively $\frac{dx_2\cdots dx_n}{e^{(n-1)\lambda}}$). For any integral function $f$ on $H^n$, using change of variable $x_1 = e^\lambda$

$$\int_{H^n} f(x_1,\ldots,x_n)\frac{dx_1\cdots dx_n}{x_1^n} = \int_\lambda \int_{(x_2,\ldots,x_n)\in\mathbf{R}^{n-1}} f(e^\lambda,x_2,\ldots,x_n)\frac{dx_2\cdots dx_n}{e^{n\lambda}}e^\lambda d\lambda$$

$$= \int_\lambda \int_{(x_2,\ldots,x_n)\in\mathbf{R}^{n-1}} f(e^\lambda,x_2,\ldots,x_n)\frac{dx_2\cdots dx_n}{e^{(n-1)\lambda}}d\lambda$$

$$= \int_\lambda \int_{\xi_{\lambda,\omega}} f(z)dVol_{\xi_{\lambda,\omega}}(z)d\lambda.$$

The above identity is equivalent to the integral formula $\int_{\mathbf{H}^n} f(x)dVol(x) = \int_{\mathbf{R}}\left(\int_{\xi_{\lambda,\omega}} f(z)dVol_{\xi_{\lambda,\omega}}(z)\right)d\lambda$. presented in (7), according to the Riemannian isometry in (A.2).

### A.5 THE HEURISTIC IS NOT A PROOF

The spectral theory does not directly lead to universal approximation theorems because of the following: 1, superpositions in (1, 2) and (8, 9) are different (similarly, although another kind of superposition in Hilbert's 13th problem (Hilbert, 1935; Arnold, 2009) was a driving force for universal approximation theorems (Nielsen, 1987), the former is hardly relevant for networks (Girosi & Poggio, 1989)); 2, desired representation properties of hyperbolic eigenfunctions are unknown, partially because $\mathbf{H}^n$ is non-compact; 3, results in spectral theory favor Hilbert spaces, while universal approximation theorems embrace more than $L^2$ space.

### A.6 OPTIMIZATION

The parameters update for the horocycle unit (2) involves the optimization problem on the sphere (for $\omega$) and the hyperbolic space (for $x$). We use a standard algorithm of sphere optimization (Absil et al., 2008) to update $\omega$, and in the supplement we present an optimization approach based on the geodesic polar-coordinates to update $x$.

In the implementation of a horocycle layer, the forward propagation is trivial, while the backpropagation involves optimization on the sphere and hyperbolic space. In the following, $\eta$ is the learning rate, $\alpha_t$ is the value of $\alpha$ ($\alpha$ may be $\eta, s, z, \omega, \ldots$) at the $t$-th step, $T_pX$ is the tangent fiber at $p$, $\nabla$ is the gradient, and $\nabla_{\mathbf{H}}$ is the hyperbolic gradient. It suffices to consider the layer $s=\langle z,\omega\rangle$.

**Optimization on the sphere**  The parameter update of $\omega$ in $s=\langle z,\omega\rangle$ involves the optimization on the sphere. The projection of $\frac{\partial\mathcal{L}_\theta}{\partial s}\nabla s(\omega_t) = \frac{\partial\mathcal{L}_\theta}{\partial s}\frac{z_t-\omega_t}{|z_t-\omega_t|^2} \in T_{\omega_t}\mathbf{R}^n$ onto $T_{\omega_t}S^{n-1}$ is given by Absil et al. (2008)[p.48]

$$v_t = \frac{\partial\mathcal{L}_\theta}{\partial s}\frac{z_t-\omega_t}{|z_t-\omega_t|^2} - \frac{\partial\mathcal{L}_\theta}{\partial s}\left(\frac{z_t-\omega_t}{|z_t-\omega_t|^2},\omega_t\right)\omega_t = \frac{\partial\mathcal{L}_\theta}{\partial s}\frac{z_t-(z_t,\omega_t)\omega_t}{|z_t-\omega_t|^2}.$$

Two well-known update algorithms of $w_t$ Absil et al. (2008)[p.76] are:

$$\omega_{t+1} = \cos(\eta_t|v_t|)\omega_t - \sin(\eta_t|v_t|)|v_t|^{-1}v_t;$$
$$\omega_{t+1} = (\omega_t - \eta_t v_t)/|\omega_t - \eta_t v_t|.$$

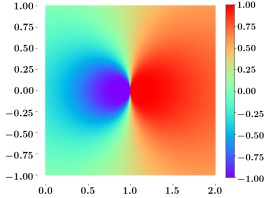

Figure 7: Poisson neuron $P^{\tanh}_{(1,0),-1/3,0}$. The level sets of a Poisson neuron $P^{\rho}_{w,\lambda,b}$ are horocycles of the ball $\{x : |x| < |w|\}$ that are tangential to $\{x : |x| = |w|\}$ at $w$ and their inverses with respect to $\{x : |x| = |w|\}$.

### A.7 A PROOF OF APOLLONIUS THEOREM

**Theorem 3** (Apollonius). *Given distinct $\omega_1, \omega_2 \in S^{n-1}$ and a positive number $\lambda$, the locus $\{x : |x - \omega_1| = \lambda|x - \omega_2|\}$ is a sphere orthogonal to $S^{n-1}$.*

*Proof.* If $\lambda$ is one then it is trivial. We assume now $\lambda$ is not one. By $|x - \omega_1| = \lambda|x - \omega_2|$, we can have

$$\left|x - \frac{\omega_1 - \lambda\omega_2}{1 - \lambda}\right|^2 = \frac{|\omega_1 - \lambda\omega_2|^2}{|1 - \lambda|^2} - 1.$$

The locus is a sphere with center $O = \frac{\omega_1 - \lambda\omega_2}{1 - \lambda}$ and radius $R = \sqrt{\frac{|\omega_1 - \lambda\omega_2|^2}{|1 - \lambda|^2} - 1}$. The theorem of Apollonius (in all dimension) claims that this sphere is orthogonal to $S^{n-1}$. To prove this, it suffices to prove $|oO|^2 = 1 + R^2$ (recall $o$ is the origin of $\mathbf{H}^n$), which follows from

$$\left|\frac{\omega_1 - \lambda\omega_2}{1 - \lambda}\right|^2 = \sqrt{\frac{|\omega_1 - \lambda\omega_2|^2}{|1 - \lambda|^2} - 1}^2 + 1.$$

$\square$

### A.8 INVERSION

On $\mathbf{R}^n \cup \{\infty\}$, given the sphere $\{x : |x - w_0| = r\}$, the corresponding inversion is given by

$$Iv(x) = w_0 + \frac{r^2(x - w_0)}{|x - w_0|^2}.$$

For $x \in \mathbf{R}^n \cup \{\infty\}$, $Iv(x)$ is called the inverse of $x$ with respect to $\{x : |x - w_0| = r\}$.

### A.9 PROOF OF THEOREM 2

**Theorem 2** *Let $K$ be a compact set in $\mathbf{H}^n$, and $1 \leq p < \infty$. Then finite sums of the form*

$$F(x) = \sum_{i=1}^{N} \alpha_i \rho(\lambda_i \langle x, \omega_i \rangle_H + b_i), \quad \omega_i \in S^{n-1}, \alpha_i, \lambda_i, b_i \in \mathbf{R}$$

*are dense in $L^p(K, \mu)$, where $\mu$ is either $d\,Vol$ (5) or the induced Euclidean volume.*

*Proof.* We first treat the case $\rho$ is sigmoidal and $\mu = d\,Vol$. Assume that these finite sums are not dense in $L^p(K, d\,Vol)$. By the Hahn-Banach theorem, there exists some nonzero $h \in L^q(K, d\,Vol)$, where $q = p/(p-1)$ if $p > 1$ and $q = \infty$ if $p = 1$, such that $\int_K F(x)h(x)d\,Vol(x) = 0$ for all finite sums of the form (14). As $K$ is a compact set, by Hölder's inequality, $\int_K |h(x)| d\,Vol \leq (\int_K d\,Vol)^{1/p} ||h||_{L^q(K, d\,Vol)}$, which leads to $h \in L^1(K, d\,Vol)$. Extend $h$ to be a function $H$

that is defined on $\mathbf{H}^n$ by assigning $H(x){=}h(x)$ if $x{\in}K$ and $H(x){=}0$ if $x{\in}\mathbf{H}^n\backslash K$. Then $H{\in}L^1(\mathbf{H}^n, d\,Vol)\cap L^q(\mathbf{H}^n, d\,Vol)$ and

$$\int_{\mathbf{H}^n} F(x)H(x)d\,Vol(x) = 0 \tag{15}$$

for all finite sums of the form (14). For any $\omega{\in}S^{n-1}$ and $\lambda, b{\in}\mathbf{R}$, we set $F_{\omega,\lambda,b}(x) = \rho(\lambda(\langle x, \omega\rangle_H {-} b))$. These functions are uniformly bounded, as $|F_{\omega,\lambda,b}(x)|{\leq}1$. Moreover,

$$\lim_{\lambda\to\infty} F_{\omega,\lambda,b}(x) = \begin{cases} 1 & if \quad \langle x, \omega\rangle_H > b, \\ 0 & if \quad \langle x, \omega\rangle_H < b. \end{cases} \tag{16}$$

According to (15), for all $\omega, \lambda, b$, we have $\int_{\mathbf{H}^n} F_{\omega,\lambda,b}(x)H(x)d\,Vol(x) = 0$. Functions $\{F_{\omega,\lambda,b}\}_{\lambda\in\mathbf{R}}$ converge pointwise as $\lambda{\to}\infty$, and they are uniformly bounded by $|H|{\in}L^1(\mathbf{H}^n, d\,Vol)$. By the bounded convergence theorem, for all $\omega{\in}S^{n-1}, b{\in}\mathbf{R}$, we have

$$\int_{\{x:\langle x, \omega\rangle_H > b\}} H(x)d\,Vol(x) = 0. \tag{17}$$

By the integral formula (7) (with notations defined there), (6) and (17), for all $b{\in}\mathbf{R}$,

$$\int_{2b}^{\infty} \left( \int_{\xi_{t,\omega}} H(z)d\,Vol_{\xi_{t,\omega}}(z) \right) dt = 0. \tag{18}$$

Taking the derivative of $\int_{2b}^{\infty} \left( \int_{\xi_{t,\omega}} H(z)d\,Vol_{\xi_{t,\omega}}(z) \right) dt$ with respect to $b$, we deduce from (18) that $\int_{\xi_{2b,\omega}} H(z)d\,Vol_{\xi_{2b,\omega}}(z) = 0$ for a.e. $b{\in}\mathbf{R}$. In other words, the integration of $H$ on a.e. $\xi \in \Xi_\omega$ is zero. This fact is valid for all $\omega{\in}S^{n-1}$. Therefore, the integration of $H$ on a.e. $\xi \in \Xi$ is zero. By the injectivity Theorem 1, $H = 0$ a.e., which contradicts our assumption. Therefore, finite sums of the form (14) are dense in $L^p(K, d\,Vol)$. The case $\rho$ is ReLU, ELU or Softplus and $\mu = d\,Vol$ follows from the above case and the fact that $x \mapsto \rho(x + 1) - \rho(x)$ is sigmoidal. The case $\mu$ is the Euclidean volume follows from previous cases and the fact that the Euclidean volume on compact $K$ is bounded from above by $\lambda d\,Vol$ for some constant $\lambda$. $\qquad\square$

### A.10 Universal approximation theorem for Poisson neurons.

In this section, $\rho$ is a continuous sigmoidal function (Cybenko, 1989), ReLU(Nair & Hinton, 2010), ELU(Clevert et al., 2016), or Softplus(Dugas et al., 2001). We also recall the Poisson neuron:

$$P^\rho_{w,\lambda,b}(x) = \rho\left( \lambda \frac{|w|^2 - |x|^2}{|x - w|^2} + b \right), \quad w \in \mathbf{R}^n, \quad \lambda, b \in \mathbf{R}.$$

**Theorem 4.** *Let $K$ be a compact set in $\mathbf{H}^n$, and $1{\leq}p{<}\infty$. Then finite sums of the form*

$$F(x) = \sum_{i=1}^{N} \alpha_i P^\rho_{\omega_i,\lambda_i,b_i}(x), \quad \omega_i{\in}S^{n-1}, \alpha_i, \lambda_i, b_i{\in}\mathbf{R} \tag{19}$$

*are dense in $L^p(K, \mu)$, where $\mu$ is either $d\,Vol$ (5) or the induced Euclidean volume.*

*Proof.* We first treat the case $\rho$ is sigmoidal and $\mu = d\,Vol$. Assume that these finite sums are not dense in $L^p(K, d\,Vol)$. By the Hahn-Banach theorem, there exists some nonzero $h{\in}L^q(K, d\,Vol)$, where $q{=}p/(p - 1)$ if $p{>}1$ and $q{=}\infty$ if $p{=}1$, such that $\int_K F(x)h(x)d\,Vol(x) = 0$ for all finite sums of the form (19). As $K$ is a compact set, by Hölder's inequality, $\int_K |h(x)|\,d\,Vol \leq (\int_K d\,Vol)^{1/p}||h||_{L^q(K,d\,Vol)}$, which leads to $h{\in}L^1(K, d\,Vol)$. Extend $h$ to be a function $H$ that is defined on $\mathbf{H}^n$ by assigning $H(x){=}h(x)$ if $x{\in}K$ and $H(x){=}0$ if $x{\in}\mathbf{H}^n\backslash K$. Then $H{\in}L^1(\mathbf{H}^n, d\,Vol)\cap L^q(\mathbf{H}^n, d\,Vol)$ and

$$\int_{\mathbf{H}^n} F(x)H(x)d\,Vol(x) = 0 \tag{20}$$

for all finite sums of the form (19). For any $\omega \in S^{n-1}$, $\lambda \in \mathbf{R}$, and $b > 0$, we set

$$F_{\omega,\lambda,b}(x) = P^{\rho}_{\omega,\lambda,-\lambda b}(x) = \rho\left(\lambda\left(\frac{1-|x|^2}{|x-\omega|^2} - b\right)\right).$$

These functions are uniformly bounded, as $|F_{\omega,\lambda,b}(x)| \leq 1$. Moreover,

$$\lim_{\lambda\to\infty} F_{\omega,\lambda,b}(x) = \begin{cases} 1 & if \quad \frac{1-|x|^2}{|x-\omega|^2} > b, \\ 0 & if \quad \frac{1-|x|^2}{|x-\omega|^2} < b. \end{cases} \tag{21}$$

According to (20), for all $\omega, \lambda, b$, we have $\int_{\mathbf{H}^n} F_{\omega,\lambda,b}(x)H(x)d\,Vol(x) = 0$. Functions $\{F_{\omega,\lambda,b}\}_{\lambda\in\mathbf{R}}$ converge pointwise as $\lambda\to\infty$, and they are uniformly bounded by $|H|\in L^1(\mathbf{H}^n, d\,Vol)$. By the bounded convergence theorem, for all $\omega\in S^{n-1}, b\in\mathbf{R}$, we have

$$\int_{\{x:\langle x,\omega\rangle_H > (\log b)/2\}} H(x)d\,Vol(x) = \int_{\left\{x:\frac{1-|x|^2}{|x-\omega|^2} > b\right\}} H(x)d\,Vol(x) = 0. \tag{22}$$

By the integral formula (7) (with notations defined there), (6) and (22), for all $b\in\mathbf{R}$,

$$\int_{\log b}^{\infty}\left(\int_{\xi_{t,\omega}} H(z)d\,Vol_{\xi_{t,\omega}}(z)\right)dt = 0. \tag{23}$$

Taking the derivative of $\int_{\log b}^{\infty}\left(\int_{\xi_{t,\omega}} H(z)d\,Vol_{\xi_{t,\omega}}(z)\right)dt$ with respect to $b$, we deduce from (23) that $\int_{\xi_{\log b,\omega}} H(z)d\,Vol_{\xi_{\log b,\omega}}(z) = 0$ for a.e. $b>0$. In other words, the integration of $H$ on a.e. $\xi \in \Xi_\omega$ is zero. This fact is valid for all $\omega\in S^{n-1}$. Therefore, the integration of $H$ on a.e. $\xi \in \Xi$ is zero. By the injectivity Theorem 1, $H = 0$ a.e., which contradicts our assumption. Therefore, finite sums of the form (19) are dense in $L^p(K, d\,Vol)$. The case $\rho$ is ReLU, ELU or Softplus and $\mu = d\,Vol$ follows from the above case and the fact that $x \mapsto \rho(x+1) - \rho(x)$ is sigmoidal. The case $\mu$ is the Euclidean volume follows from previous cases and the fact that the Euclidean volume on compact $K$ is bounded from above by $\lambda d\,Vol$ for some constant $\lambda$. $\qquad\square$

We refer the reader to the difference of (16) and (21), (17) and (22), and (18) and (23). However, basically the proofs are the same. The points are the integral formula (7), the injectivity Theorem 1 and the fact that level sets of horocycle/Poisson neurons are horocycles. Moreover, as a corollary of Theorem 4, we have

**Corollary 2.** *Let $K$ be a compact set in $\mathbf{R}^n$, and $1 \leq p < \infty$. Then finite sums of the form*

$$F(x) = \sum_{i=1}^{N} \alpha_i P^{\rho}_{w_i,\lambda_i,b_i}(x), \quad w_i\in\mathbf{R}^n, \alpha_i, \lambda_i, b_i\in\mathbf{R}$$

*are dense in $L^p(K, \mu)$, where $\mu$ is the Euclidean volume.*

*Proof.* Because $K$ is compact, there exists a positive number $R$ such that $K \subset \{x \in \mathbf{R}^n : |x| < R\}$. By the above theorem, finite sums of the form

$$F(x) = \sum_{i=1}^{N} \alpha_i P^{\rho}_{w_i,\lambda_i,b_i}(x), \quad w_i\in S^{n-1}, \alpha_i, \lambda_i, b_i\in\mathbf{R}$$

are dense in $L^p(K/R, \mu)$. Then the corollary follows from

$$P^{\rho}_{w,\lambda,b}(x) = P^{\rho}_{w/R,\lambda,b}(x/R).$$

$\qquad\square$

## A.11 PROOF OF THE LEMMA 1

Recall

$$f^1_{a,p}(x) = \frac{2|a|}{1-|p|^2} \sinh^{-1}\left(\frac{2(-p \oplus x, a)_E}{(1-|-p \oplus x|^2)|a|}\right). \tag{24}$$

The proof of Lemma 1 follows from the following direct computation.

*Proof.* Let $t \in (0,1)$. Take $p_t = t\omega$ and $a_t = -\omega$, then we have

$$-p_t \oplus x = \frac{-t(1 - 2t(\omega,x)_E + |x|^2)\omega + (1-t^2)x}{1 - 2t(\omega,x)_E + t^2|x|^2}.$$

Let $F_t(x) = \frac{2(-p_t \oplus x, a_t)_E}{(1-|-p_t \oplus x|^2)|a_t|}$, then

$$
\begin{aligned}
F_t(x) &= \frac{2(-p_t \oplus x, a_t)_E}{(1-|-p_t \oplus x|^2)|a_t|} = \frac{2\frac{t(1-2t(\omega,x)_E+|x|^2)-(1-t^2)(x,\omega)_E}{1-2t(\omega,x)_E+t^2|x|^2}}{1 - \frac{|-t(1-2t(\omega,x)_E+|x|^2)\omega+(1-t^2)x|^2}{(1-2t(\omega,x)_E+t^2|x|^2)^2}} \\
&= \frac{2t(1-2t(\omega,x)_E+t^2|x|^2)(1-2t(\omega,x)_E+|x|^2) - 2(1-t^2)(1-2t(\omega,x)_E+t^2|x|^2)(x,\omega)_E}{(1-2t(\omega,x)_E+t^2|x|^2)^2 - |-t(1-2t(\omega,x)_E+|x|^2)\omega+(1-t^2)x|^2} \\
&= A_t(x)/B_t(x),
\end{aligned}
$$

where $A_t, B_t$ are defined as the corresponding numerator and denominator. We have

$$
\begin{aligned}
A_t(x)|_{t=1} &= 2|x-\omega|^4 \\
B_t(x)|_{t=1} &= 0 \\
\partial B_t(x)/\partial t|_{t=1} &= 2|x-\omega|^2(|x|^2-1).
\end{aligned}
$$

Let $G_t(x) = \sinh^{-1}(F_t(x)) + \log\frac{1-t}{1+t}$, then

$$G_t(x) = \log\left(\frac{A_t(x)}{B_t(x)} + \sqrt{1 + \frac{A_t^2(x)}{B_t^2(x)}}\right) + \log\frac{1-t}{1+t} = \log\left(\frac{(1-t)A_t}{(1+t)B_t} + \sqrt{\frac{(1-t)^2}{(1+t)^2} + \frac{(1-t)^2 A_t^2(x)}{(1+t)^2 B_t^2(x)}}\right).$$

By L'Hôpital's rule,

$$\lim_{t<1,t\to 1} \frac{(1-t)A_t(x)}{(1+t)B_t(x)} = \frac{-A_t(x)+(1-t)A_t'(x)}{B_t(x)+(1+t)B_t'(x)}\bigg|_{t=1} = \frac{|x-\omega|^2}{2-2|x|^2}.$$

Therefore,

$$\lim_{t<1,t\to 1} G_t(x) = \log\left(\frac{|x-\omega|^2}{1-|x|^2}\right).$$

For $t < 1$, we take $p_t = t\omega, a_t = -\omega, c_t = \frac{t^2-1}{4}, d_t = \frac{1}{2}\log\frac{1+t}{1-t}$, then for all $x \in K$,

$$\lim_{t<1,t\to 1} c_t f^1_{a_t,p_t}(x) + d_t = \lim_{t<1,t\to 1} \frac{-1}{2}G_t(x) = \frac{1}{2}\log\left(\frac{1-|x|^2}{|x-\omega|^2}\right) = \langle x, \omega\rangle_H.$$

If there exists $c_1, c_2$ such that $|c_t f^1 a_t, p_t(x) + d_t|(= |G_t(x)|/2) \le c_2$ for all $t \in (c_1,1), x \in K$, then by the dominated convergence theorem, there exists $t$ such that $||c_t f^1_{a_t,p_t}(x) + d_t - \langle x,\omega\rangle_H||_{L^p(K,m)} < \epsilon$, which proves the lemma. Note that

$$
\begin{aligned}
\frac{(1-t)A_t(x)}{(1+t)B_t(x)} &= \frac{2|x-\omega|^4(1-t) + \sum_{j=1}^4 U_j(x,\omega)(1-t)^{j+1}}{-2|x-\omega|^2(|x|^2-1)(1-t)(1+t) + \sum_{l=2}^4 L_l(x,\omega)(1-t)^l(1+t)} \\
&= \frac{2|x-\omega|^4 + \sum_{j=1}^4 U_j(x,\omega)(1-t)^j}{2|x-\omega|^2(1-|x|^2)(1+t) + \sum_{l=2}^4 L_l(x,\omega)(1-t)^{l-1}(1+t)},
\end{aligned}
$$

where $U_j$ and $L_l$ are continuous functions defined on $K \times \{\omega\}$. There exist positive numbere $c_3, c_4$ and $c_1 \in (0, 1)$ such that for all $x \in K$ and $t \in (c_1, 1)$,

$$c_3 \leq 2|x - \omega|^4 \leq c_4,$$

$$c_3 \leq 2|x - \omega|^2(1 - |x|^2)(1 + t) \leq c_4,$$

$$\frac{c_3}{2} \geq |\sum_{j=1}^{4} U_j(x, \omega)(1 - t)^j|,$$

$$\frac{c_3}{2} \geq |\sum_{l=2}^{4} L_l(x, \omega)(1 - t)^{l-1}(1 + t)|.$$

Therefore, for $x \in K$ and $t \in (c_1, 1)$, we have

$$\frac{c_3}{2c_4 + c_3} \leq \frac{(1 - t)A_t(x)}{(1 + t)B_t(x)} \leq \frac{2c_4 + c_3}{c_3}.$$

This implies that for $t \in (c_1, 1)$, $G_t|_K$ and therefore $|c_t f_{a_t, p_t}^1 + d_t||_K$ are uniformly bounded, which finishes the proof of the lemma. $\square$

## A.12 THE FIRST MNIST CLASSIFIER IN 6.1

At the preprocessing stage, we compute the projection of the $28 \times 28$ input pattern on the 40 principal components and then scale them so that the scaled 40-dimensional PCA features are within the unit ball. In our network,

1. Input layer: scaled 40-dimensional PCA features;
2. First layer: 40 inputs/1000 outputs horocycle layer (tanh activation);
3. Last layer: 1000 inputs/10 outputs affine layer;
4. Loss: cross entroy loss.

Take learning rate $= 1$, learning rate decay $= 0.999$, and batch size $= 128$, and run it three times. The average test error rates after 600 epochs is $\mathbf{1.96}\%$.

PCA follows LeCun et al. (1998)(C.3), where 40 PCA is used for the quadratic network. Quadratic network has a similar structure to ours, because our neuron are contructed by quotient of quadratic functions followed by log.

## A.13 HOROCYCLE LAYER FOLLOWED BY MLR CAN APPROXIMATE THE CLASSFICATION FUNCTION

Suppose the MNIST classification function $\mathcal{M}$ is defined on $\cup_{j=0}^{9} K_j \subset \mathbf{H}^{40}$, where $K_i$ are relatively compact and $\mathcal{M}|_{K_j} = j$. By Theorem 2, for $0 \leq j \leq 9$, there exist $F_j(x) = \sum_{i=1}^{N_j} \alpha_{j,i} \rho(\lambda_{j,i} \langle x, \omega_{j,i} \rangle_H + b_{j,i})$ such that $F_j$ approximates $I_{K_j}$, where $I$ is the indicator function. Therefore, a network with the first (horocycle) layer given by $\rho(\lambda_{j,i} \langle x, \omega_{j,i} \rangle_H + b_{j,i})(0 \leq j \leq 9, 1 \leq i \leq N_j)$ followed by a classical MLR with parameters given by $\alpha_{j,i}(0 \leq j \leq 9, 1 \leq i \leq N_j)$ (with $\arg\max$ for prediction) approximates $\mathcal{M}$.

## A.14 THE SECOND MNIST CLASSIFIER IN 6.1

At the preprocessing stage, we do data augmentation by letting each image 1 step toward each of its 4 corners, so that our traning set has 300000 examples. In our network,

1. Input layer: (28,28, 1);
2. First block: 32-filters $3 \times 3$ convolution, ReLU, $2 \times 2$ max-pooling, BatchNorm;
3. Second block: 64-filters $3 \times 3$ convolution, ReLU, BatchNorm;
4. Thrid block: 64-filters $3 \times 3$ convolution,ReLU,$2 \times 2$ max-pooling, BatchNorm;

5. Fourth block: 128-filters $3 \times 3$ convolution, ReLU, $2 \times 2$ max-pooling, BatchNorm;

6. Fifth block: FC 1000, ReLU, BatchNorm;

7. Last block: 1000 input/10 output Poisson layer, sigmoid, BatchNorm;

8. Loss: cross entroy loss.

In optimization, we take Adam(Kingma & Ba, 2015). The batch size is 128 in the first 5 epochs, and 1024 in the next 15 epochs. After 5 epochs, we set $\omega_i$ in the Poisson layer to be non-trainable. We train our network five times, the average test error rate after 20 epochs is $0.35\%$.

The $\epsilon$ in $\frac{|w|^2 - |x|^2}{|x-w|^2 + \epsilon}$ is an important hyperparameter for the numerical stability. We train this MNIST model with $\epsilon \in \{10^{-1}, 10^{-2}, 10^{-4}, 10^{-6}, 10^{-8}, 10^{-10}, 10^{-20}\}$. They all show robust performance.

### A.15 Experiment of Poincare tree classification task

Given a Poincaré embedding (Nickel & Kiela, 2017) $\mathcal{PE} : \{\text{WordNet noun}\} \to \mathbf{H}^D$ of the 82114 WordNet noun nodes and given a node $x$, the task is to classify all other nodes as being part of the subtree rooted at $x$ (Ganea et al., 2018a). Our model is a logistic regression, where the horocycle feature $p \in \{\text{WordNet noun}\} \mapsto h_{\mathcal{PE}(x)}(\mathcal{PE}(p)/s)$ ($s$ is a hyperparameter lying in $[1, 1.5]$) is the only predictor, and the dependent variable is whether $p$ is in the subtree rooted at $x$. Let $P$ be the set of all nodes in the Poincare embedding, and let $p$ range from $P$.

1. Input: $h_{\mathcal{PE}(x)}(\mathcal{PE}(p)/s)$ (s is a hyperparameter.)

2. Only layer: 1 input/1 output affine layer. (two parameters: one for input, one for bias.)

3. Loss: Logistic. (with respect to 1 if $p$ in the tree rooted at $x$; 0 else.)

In each training, $x$ is one of {animal, group, location, mammal, worker}, dim is one of {2,3,5,10}, and Poincaré embeddings are from the animation_train.py of Ganea et al. (2018b) [4] (with tree=wordnet_full, model=poincare, dim=dim, seed randomly $\in \{7, 8, 9\}$). All nodes in the subtree rooted at $x$ are divided into training nodes (80%) and test nodes (20%). The same splitting procedure applies for the rest nodes. We choose $s$ that has the best training F1, and then record the corresponding test F1. For each $x$ and dim, we do the training 100 times. The average test F1 classification scores are recorded in Table 2.

The horocycle feature performs well here because it is compatible with the Poincaré embedding algorithm. Let $x$ be a node that is not at the origin. It seems that the Poincaré embedding algorithm tends to pull all nodes that are from the subtree rooted at x towards the direction of $\frac{x}{|x|}$, therefore $y \to \left\langle y, \frac{x}{|x|} \right\rangle_H$ is a suitable feature for this task.

### A.16 End-based clustering in $\mathbf{H}^2$

For MNIST, at the preprocessing stage, we do data augmentation by letting each image 1 step toward each of its 4 corners, so that our traning set has 300000 examples. Our network for $\mathbf{H}^2$ embedding of MNIST dataset is

1. Input layer: (28,28, 1);

2. First block: 32-filters $3 \times 3$ convolution, ReLU, $2 \times 2$ max-pooling, BatchNorm;

3. Second block: 64-filters $3 \times 3$ convolution, ReLU, BatchNorm;

4. Thrid block: 64-filters $3 \times 3$ convolution, ReLU, $2 \times 2$ max-pooling, BatchNorm;

5. Fourth block: 128-filters $3 \times 3$ convolution, ReLU, $2 \times 2$ max-pooling, BatchNorm;

6. Fifth block: FC 1000, ReLU, BatchNorm;

7. Sixth block: FC 2, ReLU, BatchNorm, $\mathrm{Exp}$;

8. Last block: 2 input/10 output horocycle layer, sigmoid;

---

[4]https://github.com/dalab/hyperbolic_cones

9. Loss: cross entroy loss,

where Exp is the exponential map $T_o\mathbf{H}^2(=\mathbf{R}^2) \to \mathbf{H}^2$. We apply the data augmentation as in A.14. In optimization, learning rate is 0.1, learning rate decay is 0.99, batch size is 128, epochs is 50.

Our network, data augmentation and optimization for $\mathbf{H}^2$ embedding of Fashion-MNIST dataset is completely the same as that for MNIST.

For MNIST and Fashion-MNIST we use sphere optimization. We would like to remark that there are interesting new features in sphere optimization. Because the $S^1$ is compact, for any continuous function $f$, there exists $x = argmax_{S^1} f$. The derivative of $f$ at $x$ vanish, so the usual optimization algorithm to find the minimum will fail in the general case. In our experiments, we solve this problem by adding the following tricks:

1. Observation: if the class $C_\alpha$ are all close to $\omega \in S^1$, and the end prototype $\omega_\alpha$ for the class $C_\alpha$ is around $-\omega$, then $\omega_\alpha$ is a maximum point of the loss function and therefore can not be improved through normal SGD. We solve this problem by adopting an idea(supervised variation) of k-means clustering. In each early epochs, optimization consists of two parts. In the first part, the normal SGD applies. In the second part, we move end prototypes ($\omega_i$) to the average direction of the class (using training data).

2. Observation: if the class $C_\alpha$ and class $C_\beta$ are all close to $\omega \in S^1$, and the end prototype $\omega_\alpha, \omega_\beta$ are also both around $\omega$, then all points in class $C_\alpha$ and class $C_\beta$, end prototypes $\omega_\alpha, \omega_\beta$ will all be pulling to $\omega$ by the SGD, and finally the network can not distinguish class $C_\alpha$ and class $C_\beta$. We solve this problem by adding a loss if two prototypes are close.

With these small tricks, our 2D end-based clustering algorithm is very stable for MNIST and Fashion-MNIST. We run it on MNIST 10 times, and they all get a test acc around 99% within 20 epochs.

Suppose the classification task has $M$ classes and the prototype of the $i$-th class is $\omega_i$. We write down the additional loss function for the second observation as follows

$$i = \text{RandomChoice}(\{1, \ldots, M\})$$
$$j = \text{RandomChoice}(\{1, \ldots, M\} \setminus \{i\})$$
$$d = (\omega_i, \omega_j)_E$$
$$L_{\text{Observation2}} = \text{arctanh}(10 \times \text{ReLU}(d - 0.9 - \epsilon)),$$

where $\epsilon$ is a small constant for numerical stability.

For CIFAR-10, our network for $\mathbf{H}^2$ embedding of CIFAR-10 dataset is

1. Input layer: (32,32, 3);
2. First block: ResNet-32/128 output;
3. Second block: FC 2, ReLU, BatchNorm, Exp;
4. Last block: 2 input/10 output horocycle layer;
5. Loss: cross entroy loss.

In the data augmentation, we apply horizontal/vertical shifts and horizontal flip. We use Adam. The batch size is 32 in the first 100 epochs, or 1024 in the next 50 epochs. The weights of the horocycle layer are fixed at the beginning of the training and are non-trainable, which follows an idea of Mettes et al. (2019).

## A.17 POISSON MLR

For CIFAR-10, we use a ResNet-32 structure as the feature descriptor, and we apply horizontal/vertical shifts and horizontal flip. In our network,

1. Input layer: (32,32, 3);
2. First block: ResNet-32/128 output;
3. Second block: FC 128, ReLU, BatchNorm;

4. Last block: 128 input/10 output Poisson layer, BatchNorm;

5. Loss: cross entroy loss.

We use Adam. The batch size is 32 in the first 80 epochs, or 1024 in the next 20 epochs. Test acc greater than $93.5\%$.

For the classification task of flowers (Tensorflow), The dataset of 3670 photos of flowers contains 5 classes: daisy, dandelion, roses, sunflowers and tulips. The keras model is

1. Input layer: (180,180, 3);

2. First block: 16-filters $3 \times 3$ convolution, ReLU, $2 \times 2$ max-pooling;

3. Second block: 32-filters $3 \times 3$ convolution, ReLU, $2 \times 2$ max-pooling;

4. Thrid block: 64-filters $3 \times 3$ convolution,ReLU,$2 \times 2$ max-pooling;

5. Fourth block: FC 128, ReLU;

6. Last block: 128 input/10 output FC layer;

7. Loss: cross entroy loss.

Our Poisson model is

1. Input layer: (180,180, 3);

2. First block: 16-filters $3 \times 3$ convolution, ReLU, $2 \times 2$ max-pooling;

3. Second block: 32-filters $3 \times 3$ convolution, ReLU, $2 \times 2$ max-pooling;

4. Thrid block: 64-filters $3 \times 3$ convolution,ReLU,$2 \times 2$ max-pooling;

5. Fourth block: FC 128, ReLU;

6. Last block: BatchNorm, 128 input/10 output Poisson layer, sigmoid, BatchNorm;

7. Loss: cross entroy loss.

We use 2936 photos for training and the rest 734 for testing. We train two models 5 times, and Figure 6 records the average of training and test accuracies in 20 epochs. Although the test accuracy of the Poisson model is bad in the beginning, it is fairly higher (around $4\%$) than the test accuracy of the keras model in the end.

## A.18   Dist is a relative distance function

As $\mathbf{H}^n$ is a complete metric space, the absolute distance between any pair pf points $(x, \omega) \in \mathbf{H}^n \times S^{n-1}$ is always $+\infty$. This absolute distance is not useful, hence we look for a relative one. Moreover, if a function $D$ reasonably measures the relative distance of $\mathbf{H}^n$ from $\omega \in S^{n-1}$, then so does $D + c$ for any constant $c$. This section explains why Dist (10) is a reasonable relative distance function. For $(x, \omega, b) \in \mathbf{H}^n \times S^{n-1} \times \mathbf{R}$, we recall

$$\mathrm{Dist}(x, \omega, b) = -\log\left(\frac{1 - |x|^2}{|x - \omega|^2}\right) + b = -2\langle x, \omega \rangle_H + b.$$

### A.18.1   A naive viewpoint

For fixed $\omega \in S^{n-1}$ and $b \in \mathbf{R}$, $\mathrm{Dist}(\cdot, \omega, b)$ is defined on $\mathbf{H}^n$, and its level sets are the set of horocycles that are tangential to $S^{n-1}$ at $\omega$: $\Xi_\omega = \cup_{\lambda \in \mathbf{R}}\{\xi_{\lambda,\omega}\}$. For $\lambda \in \mathbf{R}$, using (6), we have for all $x \in \xi_{\lambda,\omega}$,

$$\mathrm{Dist}(x, \omega, b) = -\lambda + b.$$

The above identity implies the following: if $x_1, x_2$ are on the same horocycle $\xi_{\lambda,\omega}$ then $\mathrm{Dist}(x_1, \omega, b) = \mathrm{Dist}(x_2, \omega, b) = -\lambda + b$. Moreover, as $\lambda$ moves to $\infty$ (equivalently speaking, the horocycle $\xi_{\lambda,\omega}$ moves to $\omega$), then for $x \in \xi_{\lambda,\omega}$ the function value $\mathrm{Dist}(x, \omega, b)$ goes to $-\infty$; as $\lambda$ moves to $-\infty$ (equivalently speaking, the $\xi_{\lambda,\omega}$ moves away from $\omega$), then for $x \in \xi_{\lambda,\omega}$ the function value $\mathrm{Dist}(x, \omega, b)$ goes to $\infty$. The above observation heuristically explains that $\mathrm{Dist}(\cdot, \omega, b)$ measures the relative distance of $\mathbf{H}^n$ from $\omega$.

### A.18.2 THE BUSEMANN FUNCTION VIEWPOINT.

Fix $\omega \in S^{n-1}$, and then let $c : [0, \infty) \to \mathbf{H}^n$ be the unique geodesic ray (with unit speed) that satisfies

$$c[0] = (0, \ldots, 0), \quad c(\infty) = \omega.$$

For the purpose of this section, we do not need the definition of Busemann functions (we refer the interested reader to Bridson & Haefliger (2009)[II.8]). Instead, it suffices to know the following result of the theory: let $d_{\mathbf{H}^n}$ be the hyperbolic distance function, and for any $x \in \mathbf{H}^n$,

$$\lim_{t \to \infty} (d_{\mathbf{H}^n}(x, c(t)) - d_{\mathbf{H}^n}(c(0), c(t))) = -2\langle x, \omega \rangle_H. \tag{25}$$

We read the above (25) in the following way: for fixed $t > 0$

$$x \in \mathbf{H}^n \mapsto d_{\mathbf{H}^n}(x, c(t)) - d_{\mathbf{H}^n}(c(0), c(t))$$

is a function that measures the relative distance of $\{x, c(0)\}$ from $c(t)$, and therefore the left hand side of (25)

$$x \in \mathbf{H}^n \mapsto \lim_{t \to \infty} (d_{\mathbf{H}^n}(x, c(t)) - d_{\mathbf{H}^n}(c(0), c(t)))$$

is a function that measures the relative distance of $\{x, c(0)\}$ from $\lim_{t \to \infty} c(t) = \omega$, and finally the right hand side of (25)

$$x \in \mathbf{H}^n \mapsto -2\langle x, \omega \rangle_H$$

is a function that measures the relative distance of $\{x, c(0)\}$ from $\omega$.

Moreover, if the geodesic ray $c$ starts from a different $y \in \mathbf{H}^n$, and then there will be a corresponding bias term added to (25). Therefore, for any $b \in \mathbf{R}$ and $\omega \in S^{n-1}$,

$$x \in \mathbf{H}^n \mapsto \mathrm{Dist}(x, \omega, b) = -2\langle x, \omega \rangle_H + b$$

is a function that measures the relative distance of $\mathbf{H}^n$ from $\omega$.

