# OpenReview forum: "Laplacian Eigenspaces, Horocycles and Neuron Models on Hyperbolic Spaces"
_ICLR.cc/2021/Conference — Reject_

### Official Review · AnonReviewer1 · 2020-10-27
**Review Comment #1**

**Rating:** 4
**Confidence:** 3

**Review:**

**Review summary**

The proposed models are theoretically sound, as it is as expressive as satisfying the universality theorem. Also, the proposed methods are empirically superior to existing methods. However, I think there is much room for improvement in the presentation of the paper. Besides, it is unclear to me what the research question is and how the proposed methods solve the problem. I would recommend reconsidering the organization of the paper.

**Summary of the paper**

This paper proposed a horocycle neuron that acts on the hyperbolic space. It uses the Hyperbolic Poisson kernel in place of the standard Euclid norm on the Euclidean space. This paper proposed an architecture called horocycle MLR, which used a horocycle neuron as a building block, and Poisson neural MLR. They showed the universality of a model with a single hidden layer of horocycle neurons or $f^1_{a, p}$, which has been used in the existing literature. They applied the horocycle feature to a subtree classification task of Poincare embedding, a horocycle MLR to a clustering task of 2D embedding, and horocycle and Poisson MLRs to classification tasks on image datasets.

**Claim**

If I understand correctly, this paper claims that the horocycle and Possison neurons are theoretically sound and empirically effective. However, it is not clear to me the research question that this paper addressed and how the theoretical and empirical properties of the proposed methods answer the question. It is true that they discussed the heuristic connection between the universal approximation property and the integral representation of the form (8) of a horocycle neuron. However, I think it is not a research question but supporting evidence that the universal approximation property is likely to hold.

**Soundness of the claims**

Can theory support the claim?
- The authors proved the universal approximation theorem for horocycle and $f^1_{a, p}$. Although it is not a constructive proof due to the Hahn-Banach theorem's nature, as the paper pointed out, it gives an affirmative answer for the theoretical justification and is a good first step to study the expressive power.
- If I do not miss any information, the Poisson neuron model (Section 4.2, Paragraph 4) is introduced without its motivation nor justification. In addition, this paper does not provide the theoretical superiority of the model. For example, Theorem 1 and Theorem 2 does not apply to the Poisson neuron model. I want to know if there are theoretical justifications for the Poisson model.

Can empirical evaluation support the claim?
- Section 6.1: I confirm that the horocycle model's overall performance is better than Ganea et al. (2018a) and Shimizu et al. (2020). Especially, the proposed method significantly outperforms them when the embedding dimension is two, or the subtree is "worker.n.01".
- Section 6.2--6.4: I confirm that the proposed method's error rate is smaller than the existing methods.
- Section 6.5: I could not understand the motivation for the experiments of the CIFAR-10 and Fashion-MNIST datasets in this section. Figure 8 claimed that Poisson MLR shows good generalization in the Flowers dataset. However, this paper does not provide such a comparison in the CIFAR-10 and Fashion-MNIST datasets. Also, the performance in these datasets is not as good as the SOTA models (I referenced [1] for CIFAR-10 and [2] for Fashion-MNIST]). Therefore, I think these results do not support the empirical superiority of Poisson MLR.
[1] https://paperswithcode.com/sota/image-classification-on-cifar-10
[2] https://paperswithcode.com/sota/image-classification-on-fashion-mnist


**Significance and novelty**

Novelty
- To the best of our knowledge, this is the first study that proves the universal approximation theorem for a single-hidden layer model on a hyperbolic space.

Relation to previous work
- Although this paper mentioned Ganea et al. (2018a) and Shimizu et al. (2020), with which this paper compare the proposed method in the experiment, it did not compare the methodological difference (especially novelty and superiority) of the proposed method from the two. Same is true of the baseline methods in Table 2 and the method by Ontrup & Ritter (2005) and Grattarola et al. (2019) in Table 3. I would like to recommend to make it clear what is the drawback of the existing model.


**Correctness**
Is the theory correct?
- Yes, So far as I check the proof, Theorem 1 and Corollary 1 (universality of horocycle neurons and the function $f^1_{a, p}$) are correct.

Is the experimental evaluation correct?
- Yes, I did not find any methodologically incorrect point in the experimental procedures. In Table 1, ideally, we should compare three methods with the same train/test partitions because the class label is highly imbalanced (e.g., 1115/82114 is positive in the case of worker.n.01 ); I am wondering if the performance variance caused by the randomness of data partition could be high.

Reproducibility of the experiments
- Yes. It explains experimental settings in detail in the appendix. Also, it has a runnable code with trained parameters.

**Clarity**

I would say that there is much improvement in the clarity of the paper.
     First, I took some time to understand how sections are related and how paragraphs in a section are related. I think adding discourse markers and organizing sentences so that readers can do paragraph reading could make the paper more understandable. Take the introduction section as an example. I feel there is a large gap between the third and fourth paragraphs. In addition, I could not understand that the fourth paragraph intends to explain the horocycle neuron until I reached the end of the paragraph. Also, although the function $f^1_{a, p}$ is introduced in the fifth paragraph, the introduction does not mention it in the remaining part and goes back to the explanation of horocycle neurons.
     Another problem is that the tables and figures are not prepared appropriately. For example, Table 3 is inserted within a paragraph. Also, captions and legends of figures are tiny and hard to read.

**Additional feedback**
- Abstract: The acronym MLR is used without what it stands for. So, I recommend writing the meaning of MLR without abbreviation.
- Section 1, Paragraph 3: This paper study → studies
- Section 1, Second bullet: Although this sentence mentioned the Poisson neuron and the horocycle MLR, they were not mentioned before this sentence. Similarly, the term "end-based" is used in the introduction but is explained in Section 4.2 for the first time. I would recommend writing their explanation before they are used.
- Section 2 Paragraph 3 (Hyperbolic deep learning): I could not see what this paper intended to mean by the term "prototype" at first reading. This wording may need some definition.
- Section 4.1, Paragraph 3 (Neuron models): What does the following sentence mean?: We accept the representation properties of eigenfunctions on compact manifolds.
- Section 4.2, Paragraph 5 (End-based clusters, end prototypes): It was hard for me to understand the relationship between sentences in the paragraph. For example, it is not clear at first sight how RBF is related to the discussion of clustering algorithms. I would recommend reconsidering the organization of the paragraph.
- Section 5 (9): $\langle x, \omega_i \rangle$ → $\langle x, \omega_i\rangle_H$
- Section 6.1, Table 1: Could you explain what $H^2$, $H^3$, etc. means?
- Section 6.1: The task (Ganea et al., 2018a) is to classify all other nodes as [...]. → It is not clear what "other" nodes mean solely from the main text. I understand it after I read the first sentence of Section A.11.
- Appendix A.15: This paper says that it adds a loss to distinguish the prototypes of 4 and 9. However, looking at the code, it seems the algorithm randomly selects two classes and adds the loss from prototypes of these classes. I want to confirm if my understanding is correct and recommend explaining the procedure if it is correct.

---

> ### Author Response · Authors · 2020-11-21
> **Initial Response**
>
> Thank you a lot for your time and effort in reading our paper and writing valuable comments. Thank you also for your interest in expressivity results and in advance for your time reading the following rebuttal.
>
> $\textbf{It is not clear to me the research question that this paper addressed
> }$
>
> $\bf{Reply:}$
>
> We studied two research problems. The first one is finding some way to generalize some universal approximation results to hyperbolic spaces. And the second question is applying spectral tools, obtained from the first problem, to existing hyperbolic learning problems.
>
> The first problem is a natural one. We answer this problem by proposing the spectral generalization of linear functions, and we prove Theorem 2.  Then the second problem also becomes a natural one: this spectral generalization of linear functions has never been used in the community, does it fit into the current hyperbolic learning? We answer this question with two results:
>
> 1, The expressivity result for $f^1_{a,p}$, which demonstrates that it applies to the field of hyperbolic neural networks [1].
>
> 2, The state-of-the-art result on the Poincare embedding subtree classification task, which proves that it applies to the field of Poincare embedding [2].
>
> $\textbf{I want to know if there are theoretical justifications for the Poisson model.
> }$
>
> $\bf{Reply:}$
>
> We confirm the answer is positive. The universal approximation theorem for Poisson models is proved in A.10 of the revision. The proof is almost in the same way as that of Theorem 2. The points are the integral formula, the injectivity Theorem 1 of Helgason, and the fact that level sets of horocycle/Poisson neurons are both related to horocycles.
>
> $\textbf{The performance of Poisson MLR in these datasets is not as good as the SOTA models.
> }$
>
> $\bf{Reply:}$
>
> Firstly, in the revision, we use a ResNet-32 and Poisson MLR. It gets an error rate of 6.46% on the test images of CIFAR-10, which is at least on par with most other methods with similar network structures (for example, compare with [3]).
>
> Secondly, in the previous version, our Poisson MLR for CIFAR-10 uses only four convolution layers and applies no data augmentation. Therefore we believe the number there is also on par with most other methods with similar network structures.
>
> Lastly, this paper is mainly about the horocycle neuron, while the Poisson neuron was supposed to support the usefulness of horocycles. We claimed three state-of-the-results in the abstract and introduction: 1, the expressivity of $f^1_{a,p}$; 2, Poincare embedding subtree classification task; 3, classification accuracy of 2-D visualization. They are all obtained by horocycle neurons but not Poisson neurons.
>
> $\textbf{I would like to recommend to make it clear what is the drawback of the existing model.
> }$
>
> $\bf{Reply:}$
>
> We compare methodological difference of experiments in the revision.
>
>
> $\textbf{Ideally, we should compare three methods with the same train/test partitions.
> }$
>
> $\bf{Reply:}$
>
> In the revision, we use the same train/test partitions and present the updated F1-scores. The performance variance is not much caused by the randomness of the data partition, but by the Poincare embedding. For example, we have three Poincare embeddings in the two-dimensional hyperbolic space. The F1 score of our model for 'location.n.01' in the first Poincare embedding is of mean 97.11 and std 0.43, in the second is of mean 97.93 and std 0.44, in the third is of mean 84.81 and std 0.87.
>
> $\textbf{Clarity
> }$
>
> $\bf{Reply:}$
>
> We tried to improve the clarity in the revision.
>
> $\textbf{How RBF is related to the discussion of clustering algorithms
> }$
>
> $\bf{Reply:}$
>
> [4] uses RBF to calculate scores, and it points out that "The output of a particular RBF can be interpreted as a penalty term measuring the fit between the input pattern and a MODEL of the class associated with the RBF." Here the MODEL is essentially the prototype. We have rewritten Section 4.2 in revision.
>
> $\textbf{Could you explain what $H^2$ means?
> }$
>
> $\bf{Reply:}$
>
> $H^d$ is the d-dimensional hyperbolic space.
>
> $\textbf{The algorithm randomly selects two classes and adds the loss from prototypes of these classes.
> }$
>
> $\bf{Reply:}$
>
> The '4' and '9' are for example. In the revision, we replace '4' and '9' by $C_{\alpha}$ and $C_{\beta}$. We confirm that the algorithm randomly selects two classes and adds the loss from prototypes of these classes. We display this part of loss in the A.16 [page 23] of the revision.
>
> We also tried improving others in the Additional feedback.
>
> $\bf{References:}$
>
> [1] Octavian-Eugen Ganea, Gary Bécigneul, Thomas Hofmann, Hyperbolic Neural Networks
>
> [2] Maximilian Nickel, Douwe Kiela, Poincaré Embeddings for Learning Hierarchical Representations
>
> [3] Hong-Ming Yang, Xu-Yao Zhang, Fei Yin, Cheng-Lin Liu, Robust Classification with Convolutional Prototype Learning
>
> [4] Y. LeCun, L. Bottou, Y. Bengio, and P. Haffner. Gradient-based learning applied to document recognition.

---

> > ### Comment · AnonReviewer1 · 2020-11-21
> > **Reply to authors' initial comments**
> >
> > I thank the authors for taking my review comments seriously. I feel that the presentation of the updated version of the paper has been improved. I appreciate the authors' work. I want to take more time to check the updated version of the paper and give the authos additional questions if any.
> >
> > # Reply to the initial comments.
> >
> > **It is not clear to me the research question that this paper addressed**
> >
> > I understand that this paper's motivation is two folds. One is to generalize the universality theorem from the Euclidean space to the hyperbolic space. The other one is to demonstrate that devised tools are useful in practical hyperbolic learning. I see that these motivations are included in the updated version of the introduction.
> >
> > **I want to know if there are theoretical justifications for the Poisson model**.
> >
> > I agree with the authors in that the universality of the Poisson model (Theorem 4) can justify it theoretically. Let me take time to confirm the correctness of Theorem 4.
> >
> > **The performance of Poisson MLR in these datasets is not as good as the SOTA models.**
> >
> > I understand from Section 6.4 in the updated paper intended to demonstrate the effectiveness of Poisson MLR by comparing ResNet 32 with Poisson MLR with prototypical learning models using ResNet 32 on the CIFAR10 dataset. The former model's error rate is 5.88%, while the latter ones are 7.50%, 7.40%, and 7.37%, respectively (according to Yang et al. (2018), Table 2). So, at first sight, it seems a good improvement. However, I want to take time if we can compare these numbers directly.
> >
> > **Ideally, we should compare three methods with the same train/test partitions.**
> >
> > I appreciate that Table 2 (Table 1 in the previous version) has standard deviations and compares models using the same dataset partition. Also, I understand that the variance of the F1 score comes not only from the data partition but also from the choice of the Poincare embedding types.
> >
> > **How RBF is related to the discussion of clustering algorithms**
> >
> > OK
> >
> > **Could you explain what H2 means?**
> >
> > OK
> >
> > **The algorithm randomly selects two classes and adds the loss from prototypes of these classes.**
> >
> > OK
> >
> > # Additional feedback
> >
> > I think it is better not to place tables and figures in the middle of paragraphs (e.g., Figures 1, 2, 4, 6) because it is hard to see borderlines between the main article and figures (+captions).

---

> > > ### Author Response · Authors · 2020-11-23
> > > **Reply to Reviewer1's "Additional feedback"**
> > >
> > > Thank you a lot for your time, reply, and additional feedback.
> > >
> > > $\textbf{It is better not to place tables and figures in the middle of paragraphs}
> > > $
> > >
> > > $\bf{Reply:}$
> > >
> > > In the updated revision, we tried to place all tables and figures on the top/bottom of the page, or at the end of the corresponding subsection.

---

### Official Review · AnonReviewer2 · 2020-10-29
**A new connection between hyperbolic geometry and deep learning**

**Rating:** 8
**Confidence:** 4

**Review:**

This paper introduced a new hyperbolic neuron based on horocycles (hyperbolic counterparts of hyperplanes). The authors proved that these neurons in H^n are as useful as traditional neurons in R^n through theoretical arguments and demonstrated they can significantly improve learning in hyperbolic embeddings of tree datasets and MNIST/CIFAR datasets.

Quality:
This contribution has both theoretical and practical strengths. Theoretically, they proved that the proposed hyperbolic neurons are universal approximators (Theorem 2). Practically, they introduced a new kind of hyperbolic neuron, with its difference with existing literature clearly demonstrated through formulations and density plots. It shows supervisor performance improvements in several examples.

Clarity:
The language is well polished. The formulations and statements are clear and consistent. The presentation has high clarity with good intuitions through illustrations.

Originality:
The proposed method is mostly related to hyperbolic neural networks constructed using Mobius arithmetic operators. Their difference is demonstrated both intuitively and empirically through experiments. The relationship with previous works is clearly stated in section 2. The references are proper with page numbers mentioned.

Significance:
This paper establishes a new connection between hyperbolic geometry and deep learning. Therefore it should be interesting to the large group of audiences in those areas.

My main concern and questions are listed as follows:

Most importantly, the introduction and the theorem are based on equation (2); while the experiments are based on the Poisson layer introduced in section 4.2. I can see some inconsistency here: clearly they are different functions. Please fill this gap in the rebuttal and next version.

Are there any explanations and technical arguments of the good empirical performance?

Clearly, the hyper-parameter epsilon is important to maintain numerical stability. There should be some demonstrations in the main text or the  supplementary material to show the robustness to instability (e.g. by setting epsilon=0)

Finally, I summarize the pros and cons as follows.

Pro:
- new hyperbolic deep learning
- proof of representation power on H^n
- strong empirical results

Con:
- missing connection between equation(2) and Poisson layer

Overall, based on the above assessment measures I recommend strong acceptance.

Here are more comments for the authors' revision:

abstract: "MLR" is the abbrev of?

Introduction: introduce notation T_p(H)

After theorem 1 there have to be some remarks to explain the statement.

Same for Lemma/Corollary 1.

Theorem 2 is referred to before the statement.

The volume element "dm" is a bit hard to read

Where are the notation h_x(y) used?

Figure 8 x-axis and y-axis are not clear

-----
After rebuttal:

Thank you for the revision and the clarifications.

It is now clear that this work actually proposed two different neurons: the horocycle neuron defined on H^n and the Position neuron defined on R^n (removing one point). After the revision, they are proved to satisfy the universal approximation property. They share similar level sets (although the density of the level sets is different). It would be interesting to see their relationships through formal arguments and a more careful empirical comparison.

This work needs background knowledge in hyperbolic geometry and may not be easy to read at the beginning. That could explain the criticism regarding clarity. Overall, I believe this paper developed important tools along the line of hyperbolic deep learning and still recommends strong acceptance.

---

> ### Author Response · Authors · 2020-11-21
> **Initial Response**
>
> Thank you a lot for your time and effort in reading our paper and writing valuable comments. Thank you also for your positive feedback and in advance for your time reading the following rebuttal.
>
> $\textbf{Most importantly, the introduction and the theorem are based on equation (2); while the experiments are based on the Poisson layer introduced in section 4.2.
> }$
>
> $\bf{Reply:}$
>
> We add motivation and write down the relationship between horocycle and Poisson neuron at the beginning of the paragraph of Poisson neuron and also prove the universal approximation theorem for Poisson neurons in A.10 of the revision.
>
> The horocycle neuron is based on $\langle \cdot, \omega \rangle_H$:  $$x \mapsto \frac{1}{2} \log{\frac{1-|x|^2}{|x-\omega|^2}}, \omega \in S^{n-1}.$$
> It is only well-defined on $\mathrm{H}^n$ because of the $\log$. Some readers might not be convinced that $\log$ is necessary and that the neuron has to be defined on hyperbolic spaces.  $\log$ makes $\langle,\rangle_H$ a hyperbolic analog of $(,)_E$, and with $\log$,$\langle \cdot, \omega \rangle_H$ is more related to metric geometry (also as demonstrated by Lemma 1). However,  one can try to remove $\log$ and construct the Poisson neuron based on
> $$x \mapsto \frac{|w|^2-|x|^2}{|x-w|^2}, w \in R^n.$$
>
> The numerator is $|w|^2-|x|^2$ rather $1-|x|^2$ because then the level sets are also horocycles of the ball $\\{x:|x|<|w|\\}$. Poisson neuron seems less related to metric geometry so far, but it is more convenient for optimization as both $x$ and $w$ are well-defined on Euclidean spaces (with a pole). Lastly, their relationship is $$\frac{|w|^2-|x|^2}{|x-w|^2} = e^{2\left\langle \frac{x}{|w|}, \frac{w}{|w|} \right\rangle_{H}}, |x|<|w|.$$
>
> $\textbf{Are there any explanations and technical arguments of the good empirical performance?
> }$
>
> $\bf{Reply:}$
>
> 1, For the Poincare embedding subtree classification task: we feel it is because the horocycle feature is compatible with the Poincare embedding algorithm. Let x be a node that is not at the origin. It seems that the Poincare embedding algorithm tends to pull all nodes that are from the subtree rooted at x towards the direction of $\frac{x}{|x|}$, therefore $y \to \left\langle y, \frac{x}{|x|}\right\rangle_H$ is a suitable feature for this task.
>
> 2, For the 2-D clustering:  [1](Figure 1) points out that the traditional CNN is good at linearly separating feature representations, but the learned feature representations are of large intra-class variations. The horocycle MLR leads to the inter-class separability in the same way (angle accounts for label difference) a traditional CNN does. At the same time, it also obtains intra-class compactness. The inter-class separability suggests that the classification performance of the horocycle MLR is on par with the traditional CNN, but the intra-class compactness implies that the horocycle MLR is better at clusterings.
>
> 3, For the Poisson MLR: In the captions of Figure 4, Figure 6 of the revision, we try to compare the Poisson MLR with the classical MLR from the viewpoint of high-confidence prediction regions ([2]), which is also related to the intra-class compactness discussed in [1]. High-confidence prediction regions of the Poisson MLR are compact sets. In contrast, high-confidence prediction regions of the classical MLR are unbounded ([2]). Therefore, at the end of the training, the feature descriptor of the Poisson MLR tends to produce feature embeddings with a small intra-class variation. According to [1](Figure 1 and discussion there),  feature representation with intra-class compactness is likely to have good generalization performance. We use this viewpoint to explain the result of the experiment of the flower classification task.
>
> $\textbf{Clearly, the hyper-parameter epsilon is important to maintain numerical stability.
> }$
>
> $\bf{Reply:}$
>
> We shall add a code file to demonstrate the robustness of $\epsilon$. We demonstrate stability by training the MNIST task with
> $\epsilon \in \\{1e-1, 1e-2, 1e-4, 1e-6, 1e-8, 1e-10, 1e-20, 0\\}.$ They all show robust performance. We tried only once, and it worked also for $\epsilon=0$, which is of course because of luck. It should be important to add $\epsilon$ in $\frac{|w|^2-|x|^2}{|x-w|^2+\epsilon}$ to ensure numerical stability. We add the remark on this stability test in A.14 of the revision.
>
> $\textbf{More comments
> }$
>
> $\bf{Reply:}$
>
> MLR is replaced by multiple linear regression;$dm$ is replaced by $d\mathcal{Vol}$; $h_x(y)$ was used in the supplement section of the Poincare embedding subtree classification task, and in the revision it appears in Section 6.2. We also tried improving others in "more comments".
>
> $\bf{References:}$
>
> [1] H.-M. Yang, X.-Y. Zhang, F.Yin, C.-L.Liu, Robust Classification with Convolutional Prototype Learning
>
> [2] M. Hein, M. Andriushchenko, and J. Bitterwolf. "Why relu networks yield high-confidence predictions
> far away from the training data and how to mitigate the problem"

---

### Official Review · AnonReviewer4 · 2020-10-29
**Obscure, intriguing, confusing, and ultimately not ready for publication**

**Rating:** 5
**Confidence:** 2

**Review:**

Summary:

This paper proposes new neural models for hyperbolic space, which unlike previous hyperbolic NN works, relies on the notion of horocycle in the Poincare disk. This novel framework has connections to spectral learnig in hyperbolic space. Representation theorems alla Cybenko for layers constructed from these neurons are presented. Finally, various experiments on clustering and classifying datasets using these neurons to generate hyperbolic embeddings are presented.

With the caveat that this paper is outside my main area of expertise, I must say that I have mixed feelings about it. On the one hand, I want to like it - the topic is quite interesting and timely, the theoretical connections are intriguing, the representation results seem quite remarkable, and the experiments seem to suggest (modulo some questions I have, see below) that this is a promising approach. On the other hand, the writing, dry exposition, utter lack of discussion or intuition for most results, and the confusing setup of the experiments make it hard to produce a confident assessment. In addition, these drawbacks probably imply that the paper might not be accessible but to a few niche in the community, and might have a very limited impact.

For the reasons above, I'm leaning towards rejection, but I think that this could be a very solid paper if: (i) the results hold, (ii) the writing and exposition is improved, and (iii) the results are better discussed and motivated .

Strengths:
* Interesting problem in a flourishing but not-yet-too-crowded corner of the representation learning literature
* Seemingly very strong theoretical results (representation theorems for neural nets in hyperbolic space)
* Seemingly very convincing experimental results, outperforming alternative methods by wide margins

Weaknesses:
* The paper needs thorough rewriting. There's various typos, confusing grammatical choices, and overall, confusing writing.
* Besides grammar, etc, the paper needs to be written with an ICLR audience in mind, most of which might not be experts in hyperbolic geometry, so more hand holding is needed.
* The paper needs restructuring. Too much space is devoted to listing prior results without further explanation or discussion (e.g. Theorem 1 - what's the importance, implication of this result?). In turn, the contribution of this paper, mostly contained in Section 4.2, could benefit from more detailed discussion and motivation. In particular, I find sentences like "Suppose this Poisson neuron is non-trainable ... " very confusing. I have no idea what this whole sentence is trying to convey.
* The results in Section 5 need more discussion. Theorem 2 at least is reminiscent of other representation theorems in the NN literature, but what is reader supposed to take away from Lemma 1 and Corollary 1? Instead of provding a full proof of Theorem 2, I would suggest deferring that to the appendix, and using the additional space to discuss the importance of all these results.
* The experiments seem quite impressive, but then again, I'm not sure whether I can gauge their soundness with confidence. There are many details about the experimental setting that are either missing or not well explained. For example:
    * Are the G/S/H models in Table 1 all directly comparable? Do they have a similar number of parameters? Similar training?
    * The reported advantage of H over G/S seems to be mostly prominent in low dimensions of the Poincare ball. I would like to see a discussion on why this is the case.
    * Given how much variance the results in Table 1 seem to have, standard deviation or error bars should be reported along with the means
    * It is not clear what exactly is meant by test error on a clustering task in Section 6.2. How are the train / test samples used?
    * Many experimental/design choices are not well justified - e.g., why is the input layer scaled down with PCA in 6.3?

Other issues:
* The notion of end prototypes seems quite interesting, but I feel like it could be better explained / elaborated on, e.g., at the end of Section 4.
* In Theorem 2, it isn't clear where K is coming into play in the definition of F (given that the density argument is on $L_p(K,\mu)$, I suppose F is only defined for $x in K$?).

---

> ### Author Response · Authors · 2020-11-20
> **Initial Response-part1**
>
> Thank you a lot for your time and effort in reading our paper and writing valuable comments. Thank you also in advance for your time reading the following rebuttal.
>
>
> $\textbf{The paper needs thorough rewriting
> }$
>
> $\bf{Reply:}$
>
> We rewrite many parts of the paper in the revision, in particular Section 4.2.
>
> $\textbf{Most of ICLR audience might not be experts in hyperbolic geometry
> }$
>
> $\bf{Reply:}$
>
> We want to point out that the implementation of our tools is very simple, which use nothing more than quadratic functions and $\log$.  All other materials from hyperbolic geometry are simply for the proof of the theorem.
>
>
> $\textbf{The paper needs restructuring. Theorem 1 - what's the importance? Suppose this Poisson neuron is non-trainable very confusing.
> }$
>
> $\bf{Reply:}$
>
> Theorem 1 is for the proof of Theorem 2.
>
> We have rewritten Section 4.2. In particular, we rewrite end-based clustering and the horocycle MLR using the language of metric learning and prototypical learning.
>
> "Suppose this Poisson neuron is non-trainable" was supposed to mean "if the parameters of the Poisson neuron is not updated during training". We have rewritten the whole section.
>
>
> $\textbf{The results in Section 5 need more discussion.  Instead of provding a full proof of Theorem 2, I would suggest deferring that to the appendix.
> }$
>
> $\bf{Reply:}$
>
> Theorem 2 is a universal approximation theorem for the horocycle neuron constructed in this paper. Corollary 1 is a universal approximation theorem for $f^1_{a,p}$ that has been widely used in previous literature ([1],[2], [3]). Lemma 1 provides a relationship between horocycle neuron constructed in this paper and $f^1_{a,p}$ used in previous literature.
>
> We add more discussion in Section 5 and defer the complete proof to the supplement.
>
> $\textbf{Are the G/S/H models in Table 1 all directly comparable? Do they have a similar number of parameters? Similar training?
> }$
>
> $\bf{Reply:}$
>
> The target is the same, but the model is different. If the dimension is D, the number of parameters of our model is 3, of [1] is 2D, of [3] is D+1. Our model has fewer parameters and is fast at training.
>
> $\textbf{The reported advantage of H over G/S seems to be mostly prominent in low dimensions
> }$
>
> $\bf{Reply:}$
>
> The number of parameters is the reason. If the dimension is 10, our model has 3, [1] has 20, and [3] has 11. Even in this case, our model still gets better results on average.
>
> $\textbf{Standard deviation or error bars should be reported along with the means
> }$
>
> $\bf{Reply:}$
>
> We present the two std in the revision, although we feel it could be misleading. We use three different Poincare embeddings in each dimension, which accounts for the most variance of the performance. For example, we take three Poincare embedding in $H^2$ . The F1 score of our model for 'location.n.01' in the first $H^2$ Poincare embedding is of mean 97.11 and std 0.43, in the second is of mean 97.93 and std 0.44, in the third is of mean 84.81 and std 0.87. As a whole, the mean is above 93 and std is now around 6. This std is not caused by our model, it is caused by a different choice of the Poincare embedding. It seems [1] and [3] use one fixed Poincare embedding for each $H^D$. Another evidence that different Poincare embeddings account for the variance: [1] reports 91.92 while [3] reports 70.59 for {root='animal',dimension=3}, both using the method of [1].
>
>
> $\textbf{In Section 6.2, how are the train/test samples used?"
> }$
>
> $\bf{Reply:}$
>
> The clustering/embedding is given by a network $NN_{\theta}:R^{D} \to H^2$ parameterized by $\theta$. The prototypes are $w_1, ..., w_{10}$. There are also bias term $b_1,...,b_{10}$. Training samples are used to find out optimal $\theta, w_i, b_j$. For any (X, label) in the test sample, we check if
>                     $$\arg\min_{j}(\mathrm{Dist}(NN_{\theta}(X), w_j, b_j))=label.$$
> where $\mathrm{Dist}$ is (10) of the revision.
>
> $\textbf{Why is the input layer scaled down with PCA in 6.3
> }$
>
> $\bf{Reply:}$
>
> Scaling is necessary, as our neuron is
>               $$x \to 0.5\log((1-|x|^2)/|x-w|^2).$$
> Without scaling, $1-|x|^2$ could be negative and the above is nan.
>
> PCA follows [4], where 40 PCA is used for the quadratic network. The quadratic network has a similar structure to ours. We shall provide a code without PCA. It is a shallow horocycle network with the scaled(scaling is always needed, otherwise $\log((1-|x|^2)/|x-w|^2)$ can be nan) 28*28 dimension data as the input. After 600 epochs, it is of test acc 97.46, underperforming the one with PCA features.
>
> $\bf{References:}$
>
> [1] O-E.Ganea, G.Bécigneul, Thomas Hofmann, Hyperbolic Neural Networks
>
> [2] E.Mathieu, C.L.Lan, C.J. Maddison, R.Tomioka, Y.W.Teh, Continuous Hierarchical Representations with Poincaré Variational Auto-Encoders
>
> [3] R.Shimizu, Y.Mukuta, T.Harada, Hyperbolic Neural Networks++
>
> [4] Y. LeCun, L. Bottou, Y. Bengio, and P. Haffner. Gradient-based learning applied to document recognition.

---

> > ### Comment · AnonReviewer4 · 2020-11-24
> > **Updates**
> >
> > I thank the authors for their answers. The updated version of the paper does indeed make some progress towards clarity, which is why I'm increasing my score (4->5), but I believe there is still a long way to go -- the paper could be written in a more transparent and engaging way.  Overall, I like the direction of this work but I think it has not reached its full potential.

---

> ### Author Response · Authors · 2020-11-20
> **Initial Response-part2**
>
> Thank you for your interest in end prototypes.
>
> $\textbf{The notion of end prototypes seems quite interesting
> }$
>
> $\bf{Reply:}$
>
> We rewrite end-prototypes and hyperbolic MLR in Section 4.2. It fits naturally into prototypical learning.
>
>
> $\textbf{In Theorem 2, it isn't clear where K is coming into play in the definition of F
> }$
>
> $\bf{Reply:}$
>
> K does not come into play in the definition of F, and F is defined over the whole hyperbolic spaces. K comes into play in the approximation. If K is non-compact, then H in the proof is not necessarily integrable. If H is not integrable, we can not apply Helgason's Theorem 1.  It is a key point that Helgason's Theorem 1 holds only for $f\in L^1(H^n)$.  In other words, if K is non-compact, we can define the class of functions defined by F, but we can not prove that it can approximate any $L^p$ function defined on K.
>
> Indeed, if $K=H^n(n>=2)$, we guess the shallow horocycle (or $f^1_{a,p}$) network is not a universal approximator in $L^p(H^n)$, because the classical neural network provides no universal approximation in $L^p(R^n)(n\ge2)$.

---

### Official Review · AnonReviewer5 · 2020-11-06
**Nice idea taken from well-known geometric tools but falls short with experiments**

**Rating:** 5
**Confidence:** 5

**Review:**

This paper develops a MLR based on hyperbolic geometry. The idea is based on well-known concept of horocycle and horospheres which are known to be hyperbolic counterpart of line and plane in Euclidean geometry (see Coxter). Then the authors show the universal approximation which kind of follows similarly from the Euclidean counterpart. In fact we can probably conject that this universal approximation holds for any manifolds with constant sectional curvature.

Strength: To the best of my knowledge, this is the first paper to deal with linear models on hyperbolic spaces by borrowing geometric tools like horocycles.



Major weakness:

The ideas are borrowed from well-known geometric tools, although this is not a weakness but the theorems closely follow Euclidean counterpart. This essentially reduces the ``````"novelty" of the paper. Moreover, the experiments are ``"synthetic", there is no motivation to use such a construction in real experiment. It will be good to see the authors discuss in which real cases we need to use such a hyperbolic MLR.

1) The work should be better motivated, for example what is the motivation of using Horocycle layer and Poisson neuron layer?
2) In section 6.2, the 2D output after 4 convolutional layers seems very less expressive, why not increase the dimension? Also what is the motivation to map it to H^2?
3) In Theorem 2, eq. 9, why the inner product is Euclidean instead of hyperbolic?
4) The universal approximation theorem in Theorem 2 almost follows from the Euclidean counterpart, e.g., see https://cbmm.mit.edu/sites/default/files/publications/CBMM-Memo-054.pdf
5) What is the additional consequence of Corollary 1 other than showing we can approximate any function, similar as Theorem 2?
6) The statement in section 6.3 stating "t is the best hyperbolic geometry related MNIST classifier" does not carry much weight, e.g., what is the motivation of using MNIST images for MLR using hyperbolic geometry?
7) There is not much point for section 6.4. In most practical cases, the 1-dimensional reduction is not meaningful as it can not carry much information.
8) Section 6.5 seems very rushed including the Fig. 8 and experiment of Flowers. This section seems more like placeholder.

---

> ### Author Response · Authors · 2020-11-20
> **Initial Response**
>
> Thank you a lot for your time and effort in reading our paper and writing valuable comments. Thank you also for your interest in geometric tools and in advance for your time reading the following rebuttal.
>
>
> $\textbf{1. The motivation of using Horocycle layer and Poisson neuron layer.
> }$
>
> $\bf{Reply:}$
>
> About the horocycle layer: In the revision, we provide two theoretical justifications for the horocycle layer:
>
> 1, The universal approximation property.
>
> 2, It fits into metric learning.
>
> We present three applications of horocycle neuron/layer/MLR:
>
> 1, Prove universal approximation theorem for $f^1_{a,p}$  ([2],[3],[4]) by the property of horocycle, which suggests that horocycle fits into existing hyperbolic neural networks.
>
> 2, Perform well on the Poincare embedding tree classification task, which suggests that horocycle also fits into the theory of Poincare embedding ([5]).
>
> 3, End-based clustering.
>
> About the Poisson neuron layer:  In the revision, we add/discuss two theoretical justifications:
>
> 1, A universal approximation theorem in A.10;
>
> 2, Poisson MLR has compact high-confidence prediction regions, which is better for generalization ([6,7]).
>
> $\textbf{2. 2D output after 4 convolutional layers seems very less expressive
> }$
>
> $\bf{Reply:}$
>
> Yes, it is very less expressive, in particular for the CIFAR-10 task. The depth is the key ([1]). We use a ResNet-32 structure in the revision for the CIFAR-10 task, and the updated result is a clustering of CIFAR-10 in $H^2$ with a test error rate of $8.42\\%$.
>
> Mapping $R^d$ to $H^d$ is a popular trick in hyperbolic learning ([8]). In our case, we use it to get feature representations with small intra-class variations, which is good for clustering and robust classification (see Figure 1 and discussion of [6]).
>
> $\textbf{3. In Theorem 2, eq. 9, why the inner product is Euclidean instead of hyperbolic?
> }$
>
> $\bf{Reply:}$
>
> It should be $\langle \rangle_H$ and is revised.
>
> $\textbf{4. The universal approximation theorem  almost follows from the Euclidean counterpart [1]
> }$
>
> $\bf{Reply:}$
>
> We feel that the Gaussian/RBF network, quantitative results, and benefits of depth discussed in [1] are interesting and completely open in hyperbolic learning. So far, we have been unable to find out the evidence that the results in the [1] or other literature apply to horocycle neurons or $f^1_{a,p}$. We tried the following: if there exists a unique chart $\phi:K \subset H^d \to R^d$ such that for all linear function $f$ defined on $\phi(K)$, $f\circ \phi$ is a horocycle neuron (respectively $f^1_{a,p}$), then the universal approximation property for horocycle neurons (respectively $f^1_{a,p}$) might almost follow from the Euclidean counterpart. However, the existence of such chart is not the case.
>
> $\textbf{5. What is the additional consequence of Corollary 1.
> }$
>
> $\bf{Reply:}$
>
> Corollary 1 is about the shallow network constructed by $f^1_{a,p}$. $f^1_{a,p}$ and its variants are widely used in previously literature ([2], [3], [4]).  One consequence of Corollary 1 is that our result provides novel insights into the existing hyperbolic neural network.
>
>
> $\textbf{6.  What is the motivation of using MNIST images for MLR using hyperbolic geometry?
> }$
>
> $\bf{Reply:}$
>
> Hyperbolic space is an alternative for embedding image data ([3],[8], and this paper). Figure 5 in the revision proves that the result can be good.
>
> $\textbf{7. The 1-dimensional reduction is not meaningful as it can not carry much information.
> }$
>
> Firstly,  in the Poincare tree classification task, we use a single feature and achieve state-of-the-art. It suggests that sometimes 1-D feature can carry much information.
>
> Secondly, we agree that the 1-dimensional reduction is not that interesting at this moment, so we would like to take this subsection out.
>
> Lastly, in the abstract and introduction, we claim three state-of-the-art results: 1, the expressivity of $f^1_{a,p}$; 2, Poincare embedding subtree classification; 3, the classification accuracy of 2-D visualization. Taking this subsection out should not affect our claim.
>
> $\bf{References:}$
>
> [1] H.Mhaskar, T.Poggio. Deep vs. shallow networks: An approximation theory perspective.
>
> [2] O.-E.Ganea, G.Bécigneul, T.Hofmann, "Hyperbolic Neural Networks".
>
> [3] E.Mathieu, C.L.Lan, C.J. Maddison, R.Tomioka, Y.W.Teh, "Continuous Hierarchical Representations with Poincaré Variational Auto-Encoders".
>
> [4] R.Shimizu, Y.Mukuta, T.Harada, "Hyperbolic Neural Networks++"
>
> [5] M.Nickel, D.Kiela, "Poincaré Embeddings for Learning Hierarchical Representations".
>
> [6] H.-M.Yang, X.-Y.Zhang, F.Yin, C.-L.Liu, "Robust Classification with Convolutional Prototype Learning".
>
> [7] M. Hein, M. Andriushchenko, and J. Bitterwolf. "Why relu networks yield high-confidence predictions
> far away from the training data and how to mitigate the problem."
>
> [8] V.Khrulkov, L.Mirvakhabova, E.Ustinova, I.Oseledets, V.Lempitsky, "Hyperbolic Image Embeddings"

---

### Decision · Program_Chairs · 2021-01-07
**Final Decision**

**Decision:**

Reject

**Comment:**

Reviewers generally appreciate the contributions of the paper, namely the horocycle neuron, Poisson neuron, and the universal approximation properties. However, there are concerns, especially by R4 and R5, that the presentation is confusing, lacks clarity, and should be substantially improved.

Note: Theorem 1.7 in (Helgason, 1970) is proved explicitly for the case n=2, not for general n as claimed in (9).  Thus the Laplacian eigenspace motivation needs to be re-written/re-examined.